# The functional anatomy of elephant trunk whiskers

Nora Deiringer [1], Undine Schneeweiß[1], Lena V. Kaufmann [1,2], Lennart Eigen [1], Celina Speissegger[1], Ben Gerhardt[1], Susanne Holtze [3], Guido Fritsch[3], Frank Göritz [3], Rolf Becker[4], Andreas Ochs[4], Thomas Hildebrandt[3] & Michael Brecht [1,5✉]

Behavior and innervation suggest a high tactile sensitivity of elephant trunks. To clarify the tactile trunk periphery we studied whiskers with the following findings. Whisker density is high at the trunk tip and African savanna elephants have more trunk tip whiskers than Asian elephants. Adult elephants show striking lateralized whisker abrasion caused by lateralized trunk behavior. Elephant whiskers are thick and show little tapering. Whisker follicles are large, lack a ring sinus and their organization varies across the trunk. Follicles are innervated by ~90 axons from multiple nerves. Because elephants don't whisk, trunk movements determine whisker contacts. Whisker-arrays on the ventral trunk-ridge contact objects balanced on the ventral trunk. Trunk whiskers differ from the mobile, thin and tapered facial whiskers that sample peri-rostrum space symmetrically in many mammals. We suggest their distinctive features—being thick, non-tapered, lateralized and arranged in specific high-density arrays—evolved along with the manipulative capacities of the trunk.

[1] Bernstein Center for Computational Neuroscience Berlin, Humboldt-Universität zu Berlin, Philippstr. 13, Haus 6, 10115 Berlin, Germany. [2] Berlin School of Mind and Brain, Humboldt-Universität zu Berlin, Berlin, Germany. [3] Leibniz Institute for Zoo and Wildlife Research, Alfred-Kowalke-Strasse 17, D-10315 Berlin, Germany. [4] Berlin Zoological Garden, Hardenbergplatz 9, 10623 Berlin, Germany. [5] NeuroCure Cluster of Excellence, Humboldt-Universität zu Berlin, Berlin, Germany. ✉email: michael.brecht@bccn-berlin.de

Elephants almost constantly engage their trunk and frequently contact their environment with their trunk tip. Behavioral experiments revealed high sensitivity of the trunk tip to sensory stimuli[1]. It is therefore not surprising that the trunk receives a dense tactile innervation. Specifically, the infraorbital nerves and the trigeminal ganglia weigh about 1.5 kg in elephant cows. Accordingly, the infraorbital nerve, which provides the tactile sensory innervation of the elephant trunk, is much thicker than the elephant's optic nerve, mediating vision, and the vestibulocochlear nerve, responsible for auditory perception[2]. Despite the obvious behavioral relevance, our knowledge of the elephant's peripheral tactile specializations is limited.

The elephant trunk is a derived facial structure. It is a fusion organ that develops by the merging of a dramatically elongated nose and the upper lip[3]. This fusion occurs at late fetal stages and early elephant fetuses simply have long unfused noses[4]. The immense size and weight of the elephant trunk impose strain on the facial bones. The need to provide space for the attachment of the trunk musculature has led to an extraordinary enlargement of the elephant skull in evolution. The trunk is thought to act as a so-called muscular hydrostat. Trunk musculature consists of ~40,000 muscles[5], which compares to only 600–700 muscles in the human body. The trunk musculature is innervated by a large facial nucleus[6] with ~54,000 (Asian elephants) and 63,000 (African elephants) motor neurons[7]. Much like human grasping and hand use, elephant trunk actions are skillful and strongly lateralized[8–11].

There are three extant elephant species: The African savanna elephant (*Loxodonta africana*), the African forest elephant (*Loxodonta cyclotis*) and the Asian elephant (*Elephas maximus*). Our study focuses on Asian elephants and African savanna elephants, here referred to as African elephants. African and Asian elephants differ in trunk morphology. African elephants have two triangular protrusions at their trunk tip, so-called dorsal and ventral fingers, whereas Asian elephants have only one dorsal finger. Interestingly, such morphological differences between African and Asian elephants match with species-specific differences in trunk use—African elephants tend to pinch objects with their two fingers and Asian elephants tend to grasp/wrap objects with their trunk[12].

The tactile specializations of elephants are not well characterized. To our knowledge, the first description of whiskers on the elephant's trunk was provided by Fred Smith in 1890. In his anatomical study of the elephant skin, he differentiated between two types of hair on the elephant's skin, normal hair and bristles and gave a brief anatomical description of the follicle of those bristles. He further states that the largest follicles can be found in the trunk region[13]. In their book about the anatomy of the elephant's head, the authors Boas and Paulli confirm, that the hair on the lower lip and some of the hairs on the trunk can be regarded as real whiskers[3]. Whiskers or vibrissae are distinct from normal hair by having a blood sinus associated with their follicle. Sprinz published a report of nerve dissections on an Asian elephant trunk combined with behavioral and anatomical observations on living African and Asian elephants. He states that a majority of the nerves he traced ended in the follicles of the whiskers of the ventral trunk tip and concludes, that the whiskers are the main structure for the transmission of tactile stimuli in the trunk. From his observations on living elephants, he inferred that African elephants seem to have more whiskers on the trunk than Asian elephants and that the elephants react strongly to the whiskers being touched[14]. A landmark study by Rasmussen and Munger (1996) described the sensory neuro-histology of an Asian elephant trunk finger, finding various nerve endings and whisker patterns[15]. The whiskers of a number of closely related species, such as hyraxes[16] and manatees[17,18] have been studied in detail.

Our study is the first one to focus on elephant trunk whiskers. Our histological analysis of elephant follicles followed the pioneering work of Ebara et al. on rats and cats[19] and aimed at elucidating structure-function characteristics of elephant whisker follicles. Specifically, we asked the following questions: 1. Do the trunk whiskers of African and Asian elephants differ? 2. How many trunk whiskers are there? 3. What is the length distribution of elephant whiskers and what is the whisker thickness and geometry? 4. How are elephant whisker follicles organized? 5. How are elephant whisker follicles innervated? 6. Do elephant whiskers whisk? 7. What might be the function of specific trunk whisker arrays?

We find marked differences in trunk whiskers between African and Asian elephants. Both species have numerous thick whiskers and their length is determined by usage/abrasion. As a result, adult elephant whiskers show striking lateralization. Elephant whisker follicles differ with species and trunk region and elephants do not whisk. We conclude that elephant whisker patterns are shaped by trunk behavior and differ markedly from other mammalian facial whiskers.

## Results

**Trunk tip whiskers differ between African and Asian elephants**. A frontal view of an African and Asian elephant trunk tip is shown in Fig. 1a, b, respectively. As visible in the insets (Fig. 1a, b) elephant whiskers protrude from skin folds. Whiskers are more prominent in African than in Asian elephants, as also evident in side views of African (Fig. 1c) and Asian (Fig. 1d) trunk tips. Figure 1e, f gives a more quantitative assessment of the differences in whisker count and distribution of the two species studied. Specifically, we plotted the position of whiskers and whisker density maps for the same trunk samples shown in Fig. 1a–d. Compared to Asian elephants, African elephants have a higher whisker density and count on the inside of the tip (pinching zone), and also on the lateral and dorsal regions of the trunk tip. In both species, whisker density on the frontal part of the tip is higher than on the respective lateral parts of the trunk tip. An exception is the most distal part of the finger, which is a zone of markedly low whisker density in both species (Fig. 1e, f). We note that while pinching is more common in African elephants, both African and Asian elephants pinch objects with their finger tip. The overall whisker number on the trunk tip is significantly higher in African (621 ± 91, mean ± sd) than in Asian elephants (367 ± 44, mean ± sd) ($p = 0.0007$, Welch's $t$-test) (Fig. 1g). We observed no whiskers on the inside of the nostril proximal to the beginning of the nasal septum. More photographs of elephant trunk tips are provided in Supplementary Fig. 1. We conclude that African elephants have more and more prominent trunk tip whiskers than Asian elephants.

**Elephant trunk whisker length is lateralized and use-dependent**. Unlike facial whisker patterns of other mammals, both African (Fig. 2a) and Asian (Fig. 2b) adult elephants show striking whisker lateralization. Whiskers on one side of the trunk are longer than on the other side. Such whisker lateralization was not observed in trunk tips of newborn African (Fig. 2c) or Asian (Fig. 2d) elephants. Specifically, we observed such lateralization of whisker length in all adult elephant trunks (Fig. 2e upper), but in none of the newborn trunks studied (Fig. 2e lower); the difference between the occurrences of asymmetrical whiskers in adults and symmetrical whiskers in baby elephants is significant ($p = 0.0004$; Fisher's exact test, data from Asian and African elephants pooled). Trunk whisker lateralization is almost certainly related to lateralization in elephant trunk use. Specifically, we suggest lateralization of whisker length is caused by wear associated with

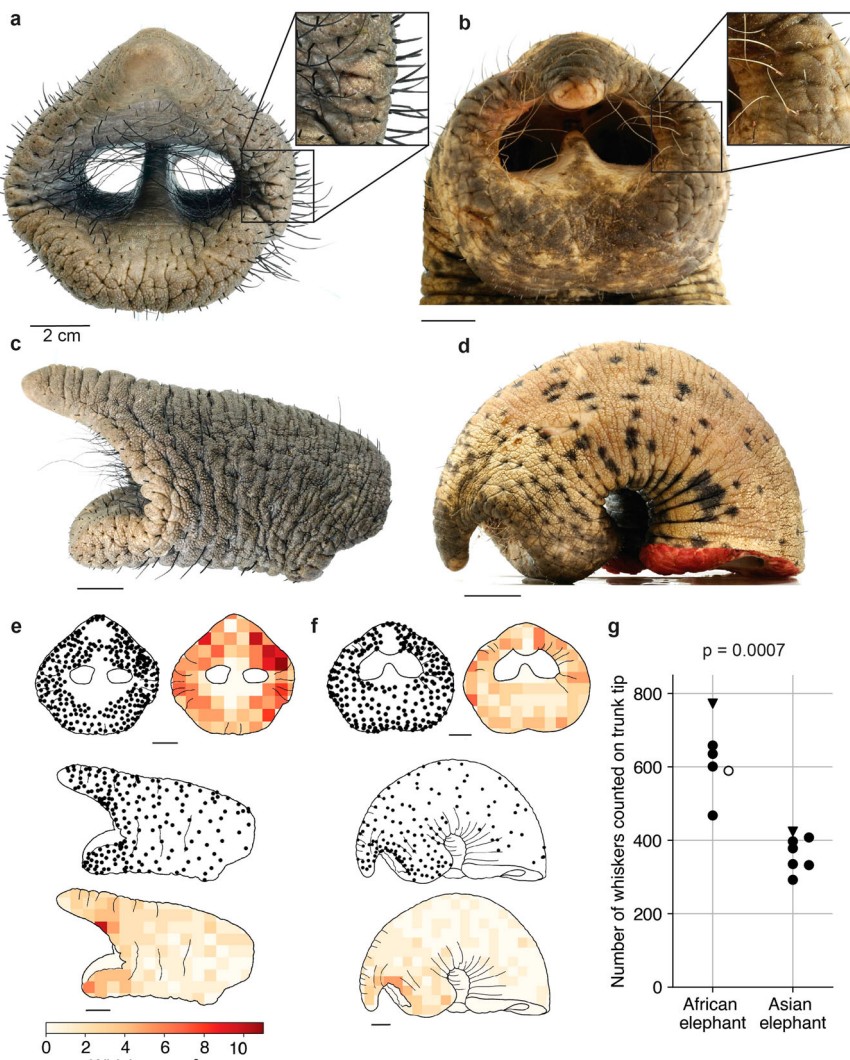

**Fig. 1 Trunk tip whisker number and density differ between African and Asian elephants. a** Frontal view of an adult African elephant cow trunk tip (Linda, see Table 1). Whiskers protrude from skinfolds (inset). **b** Frontal view of an Asian elephant trunk tip (Unknown Asian, see Table 1). Whiskers protrude from skinfolds (inset). **c** Side view of an African elephant trunk tip. **d** Side view of an Asian elephant trunk tip. **e** Upper left, dot display of whisker positions of an African elephant trunk tip from a frontal view. Upper right, whisker density (whiskers per $cm^2$) of an African elephant trunk tip from a frontal view. Middle, dot display of whisker positions of an African elephant trunk tip from a side view. Lower, whisker density of an African elephant trunk tip from a side view. **f** Whisker positions and whisker density on an Asian elephant trunk tip, conventions as in **e**. **g** Number of whiskers counted on the trunk tip of six African and seven Asian elephants. Dots (adults); triangles (newborns). One count was partially extrapolated due to an incomplete sample (empty dot). $p = 0.0007$, Welch's $t$-test (Hedges' g = 3.59).

lateralized trunk use. While we did not formally study the lateralization of elephant trunk usage here, ad hoc observations on zoo elephants supported this idea. We find that the Berlin Zoo elephant Anchali, who showed marked 'left-trunker' behavior, has shorter whiskers on the right side of the trunk. This abrasion pattern is expected in 'left-trunker' elephants, whose right trunk side touches the ground during grasping (Supplementary Fig. 2). In some regions, especially in the ventral tip area of Asian elephants, the whiskers of adult elephants are extremely short from abrasion. In contrast, both African (Fig. 2c) and Asian (Fig. 2d) newborn elephants have symmetrical and longer whiskers on the trunk. We also find that whisker replacement appears to differ between elephants and rats. We never observe dual whisker follicles in elephants by either surface inspection of the skin or in microCT scans, an observation that is common in other studies on rats (Fig. 2f)[20]. It appears, that the dual whisker regrowth pattern, that replaces entire whiskers in rats, does not apply to elephants.

**Elephant whiskers are cylindrical and differ in thickness between species.** We further studied the shape and thickness of elephant whiskers. Figure 3a shows a photograph of lateral African and Asian elephant trunk whiskers and a rat δ-whisker. We picked the rat δ-whisker for comparison, because it is a relatively long whisker in rats, in length not very different from elephant trunk whiskers. Elephant whiskers are substantially thicker and show very little tapering compared to the rat whisker, which has a conical shape. To assess whisker shape more quantitatively, we obtained microCT scans of iodine-stained whiskers of both elephants and rats. As shown in Fig. 3b–d, volume renderings of whisker base and tip, tapering and thickness differs notably between African and Asian elephants and the rat. The thickness of elephant trunk whiskers also differs between trunk regions and species. Lateral whiskers of African elephants are notably thicker than whiskers of all other regions of the trunk tip (Fig. 3e) (One way ANOVA, $P < 0.001$, pairwise comparison of lateral trunk tip whiskers with whiskers from other regions of the

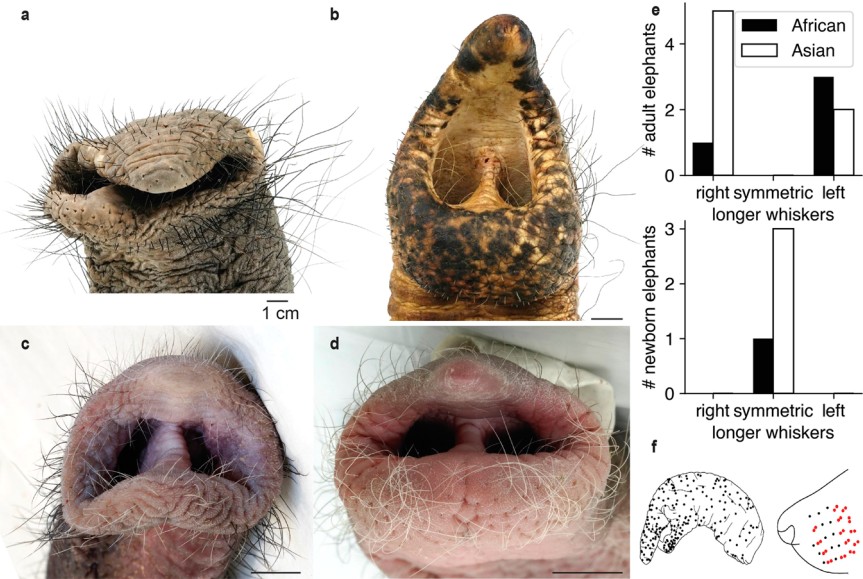

**Fig. 2 Whisker lengths varies with usage/ abrasion and is lateralized in adult elephants. a** Frontal view of an adult African elephant trunk tip. Note the asymmetric whisker length. **b** As in **a** but for an adult Asian elephant trunk tip. Note the asymmetry. **c** Frontal view of a newborn African elephant trunk tip. Note the symmetric whisker length. **d** As in **c** but for a newborn Asian elephant trunk tip. Note the symmetry. **e** Number of African (black) and Asian (white) elephant trunk tips with symmetrical whisker lengths or longer whiskers on the right or left trunk side. Upper, adult elephants. Lower, newborn elephants. **f** Left, dot display of whisker positions on the side of an African elephant trunk tip. Right, dot display of whisker positions in a rat whisker pad. Single whiskers are marked in black, double whiskers in red as reported by Maier & Brecht (2018). No double whiskers were observed in elephants.

trunk tip using a Scheffé test: $P < 0.001$ for all comparisons made), while whisker thickness in different regions of Asian elephant trunk tips differs only slightly (Fig. 3f). African elephants have thicker whiskers on the trunk tip and the lateral and ventral regions of the trunk than Asian elephants. We conclude that elephant whiskers are cylindrical, thick and sturdy.

**Whisker follicles have distinct anatomical features and differ in size**. We compared the anatomy of the follicle-sinus-complex (FSC) in African and Asian elephants and rats. To this end, we obtained microCT scans of whole iodine-stained FSCs of lateral whiskers of adult elephants and an adult rat δ-whisker. Figure 4a shows a virtual longitudinal microCT section through an African elephant whisker follicle. African elephant FSCs are more slender than the belly-shaped FSCs of Asian elephants (Fig. 4b). Both elephant follicles are much larger than the rat FSC (Fig. 4c). Histological stainings of the elephant trunk FSCs are shown in Fig. 4d–g. Like the FSCs of other species[16,21], the hair shaft of elephant trunk FSCs is surrounded by an internal and external root sheath. Connective tissue surrounds the external root sheath and connects to the collagenous capsule of the FSC on the base of the follicle. The sinus complex of elephant trunk FSCs consists only of a cavernous part, with trabeculae made from connective tissue spanning through the sinus resulting in 'vascular sinus spaces'[15]. The elephant FSCs reach into the musculature of the trunk but seem to not have associated vibrissal capsular muscles used for active whisking. To characterize differences in FSC morphology depending on the trunk region, we segmented FSCs using a microCT scan of an iodine-stained newborn Asian elephant trunk (Hoa's Baby, see Table 1). Figure 4h shows a volume rendering of the trunk and one of the segmentation sides with the segmented FSCs color-coded according to their lengths. We find that lateral FSCs are significantly longer than FSCs from tip and ventral regions of the trunk (see Fig. 4i; One way ANOVA, $P < 0.001$, post hoc Scheffé tests for pairwise comparison (*$P < 0.05$, ***$P < 0.001$)).

In summary, we show that elephant trunk whisker follicles are very large in comparison to rat whisker follicles and show marked differences in lengths according to trunk region.

**Newborn elephant trunk FSCs have high innervation**. Elephant trunk FSCs receive extensive innervation. We investigated the overall distribution of the innervation within the follicle and characterized the sensory nerve endings using immunohistochemistry. Figure 5a shows a longitudinal section through a newborn Asian elephant whisker follicle stained for Neurofilament H, a marker expressed in most sensory afferents. Two separate nerve bundles enter the FSC opposite to each other at the lower third level and split up in smaller bundles while ascending the follicle, converging closer to the hair shaft. At the upper third of the follicle, the sensory afferents evenly distribute around the circumference of the whisker and terminate between the most outer layer of the outer root sheath and mesenchymal sheath (Fig. 5a, b). Most of the follicles investigated had two or more nerve bundles penetrating the collagenous capsule of the FSC at different levels. Figure 5c shows a longitudinal section of an Asian adult whisker follicle stained with hematoxylin-eosin. The arrows indicate three nerve bundles entering the follicle, two at the lower and one at the upper level. Figure 5d shows axon numbers of FSCs from four different trunk areas of one newborn Asian elephant (Hoa's baby, see Table 1). The axon number ranges from 49 to 190 with a mean of 87. Axon numbers across follicles from the same trunk area are similar. Very small nerve bundles entering mainly at the apical part of the follicle (superficial vibrissal nerves) are not included in the count. We also investigated the presence and distribution of different mechanosensory nerve endings within the FSC. Most prominent among the nerve endings are lanceolate endings, which have an elongated, spindle-shaped structure and are situated in between the mesenchymal sheath and outer root sheath (Fig. 5e). We also observe lanceolate-like endings, that have a more droplet-like appearance (Fig. 5f). In addition, we see free nerve endings at all levels of the follicles and weakly stained endings that resemble reticular

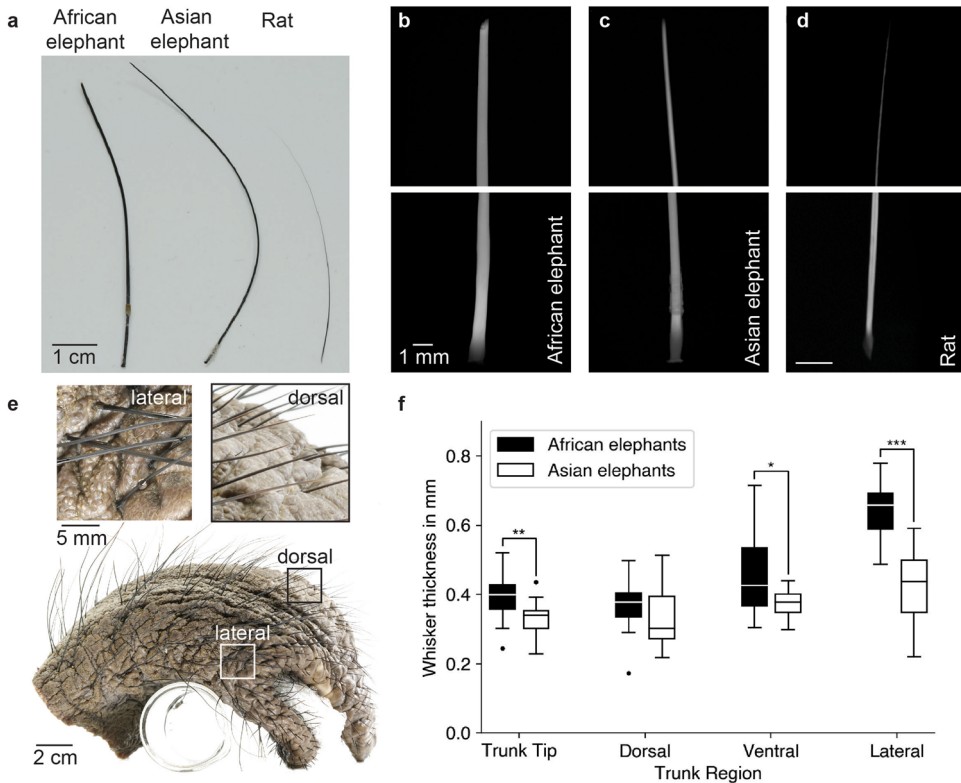

**Fig. 3 Elephant trunk whiskers are thick and show little tapering. a** Photograph of an African elephant lateral trunk whisker (left), Asian elephant lateral trunk whisker (middle) and rat δ- whisker (right). **b** Upper, volume rendering of a microCT scan showing the distal part of an iodide-stained African elephant whisker from the lateral trunk tip. Lower, proximal part of the whisker. Note the similar thickness distally and proximally. **c** Asian elephant whisker sampled on the lateral trunk tip, conventions and scaling as in **b**. Note the similar thickness distally and proximally. **d** A rat whisker (δ), conventions as in **b**. Note the tapering. **e** Side view of an African elephant trunk tip. Insets show the difference in whisker thickness between lateral and dorsal trunk tip regions. **f** Whisker thickness in different regions of the trunk tip of African and Asian elephants. While in Asian elephants, whiskers of different trunk tip regions do not differ in thickness, in African elephants, lateral whiskers are thicker than whiskers from other regions of the trunk tip. Thickness was measured from two adult Asian and two adult African elephants. A comparison of whisker thickness was done using a Welch's *t*-test (*$P < 0.05$, **$P < 0.01$, ***$P < 0.001$). Central line, median; box limits, upper and lower quartiles; whiskers, 1.5x interquartile range; dots, outliers.

endings of other species on the lower level of the follicle. Unlike in other mammalian whisker follicles[21], we observe no zone of transverse afferents and endings around the elephant whisker.

**Trunk whiskers show no active movement during haptic pinching and vacuuming.** Active tactile exploration through sweeping motions of the whiskers (whisking) is known in rodents and other mammals. We therefore asked, if elephant trunk whiskers show active movement or if whisker contact is purely determined by trunk movements. We intended to study this question in a behavioral context, in which elephants make use of haptic information and in which we could apply close-up videography. To this end, we trained the female Asian elephant Anchali in the Berlin Zoo to retrieve fruit from a box (which prevented visual control of grasping) with her trunk. While the experimental conditions were chosen such that Anchali could not guide her movements visually, we do not know which olfactory and haptic cues guided the behavior and to what extent whiskers were involved. We then collected high-speed (100 Hz frame rate) videography during pinching and vacuuming behaviors. The closed box and the experimental setup are depicted in Fig. 6a, b. Inspection of all videos collected suggests that Anchali shows no active whisker movement (Supplementary Movie 1). To document this point quantitatively we used video clips of grasping events and tracked the tip and the base of single whiskers over a time course of 1 s. This time scale allowed us to track whisker motion with negligible trunk rotation. We first tracked a lateral

trunk whisker during an instance of carrot pinching (See Fig. 6c, d for the start and the end frame of a clip). The tracked whisker along with the trajectories of the tip and the base positions during the tracking time is shown in Fig. 6e. We defined the angle between the vertical and the whisker (see Fig. 6e) as a parameter for whisker movement relative to the trunk. As shown in Fig. 6f, no whisker movement can be observed during the tracking time. Furthermore, we investigated instances of apple vacuuming in the same manner (Fig. 6g–j). This behavior is of special interest, as breathing and whisker movement are synchronized in other species such as rats[22]. We identified the onset of the inhalation/suction using the audio track of the video. In Fig. 6j we show the angle of a lateral trunk whisker relative to the vertical with the time of the inhale to create the vacuum marked in blue. No whisker movement relative to the trunk can be observed. We conclude that there was no whisking associated with haptically controlled grasping or vacuuming in our experiments.

**Ventral trunk ridge whisker arrays show behavioral contact patterns.** We obtained whole trunk whisker counts from the newborn elephant trunks shown in Fig. 7a–d. The newborn African elephant has 1220 whiskers in total, whereas the newborn Asian elephant has 986. We can observe patterns of whisker organization on the ventral side of the elephant trunk. In both African and Asian elephants whiskers are organized in two distinct rows on each side of the ventral trunk (Fig. 7a, b). However, in African elephants, these rows run through the whole length of

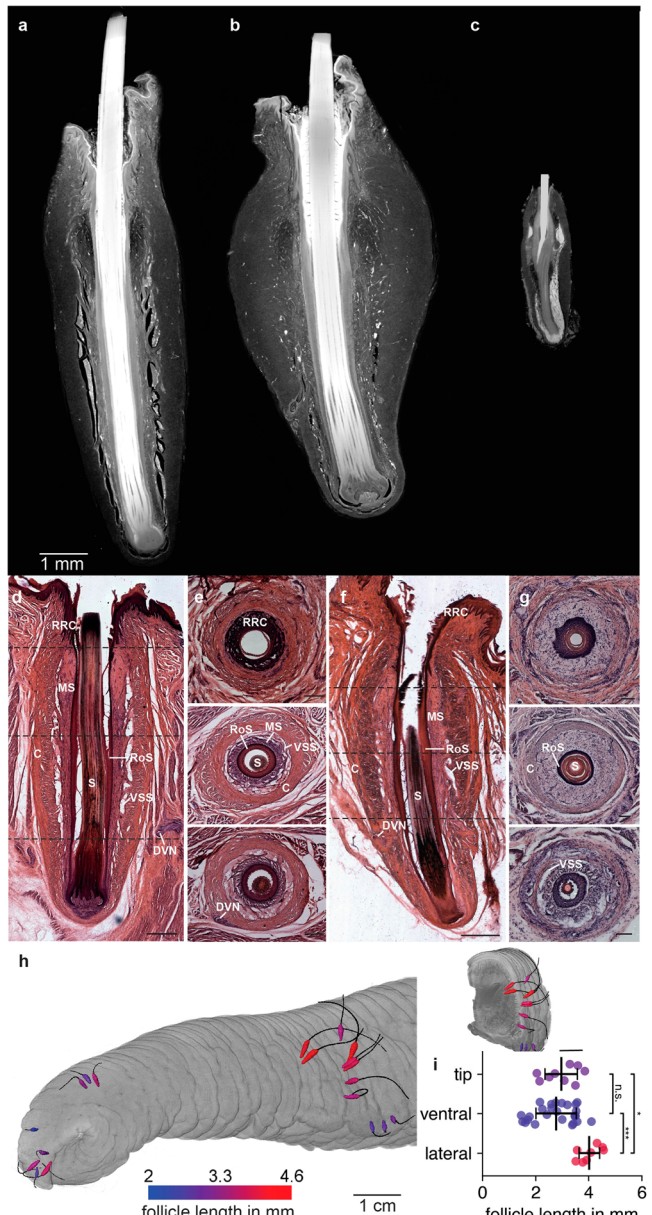

**Fig. 4 Organization of elephant trunk whisker follicles.** Virtual longitudinal microCT section of **a** a dorsal African elephant trunk follicle-sinus-complex (FSC). The follicle was stained in 1% iodine for 48 h. The surrounding tissue was cropped, and whisker contrast was selectively adjusted in some areas to maintain visibility and avoid saturation. **b** a lateral Asian elephant trunk FSC, conventions as in **a**. **c** a rat δ whisker FSC. The FSC was stained in 1% iodine for 24 h. **d** Micrograph of a hematoxylin-eosin stained longitudinal section of a lateral adult African elephant trunk FSC (scale bar = 1 mm). **e** Micrograph of hematoxylin-eosin stained transversal sections of a lateral adult African elephant trunk FSC. Approximate planes of the sections are indicated in **d** with dotted lines (scale bar = 500 μm). **f** As **d** but for a lateral adult Asian elephant trunk FSC (scale bar = 1 mm). Note that the whisker partially broke during cryosectioning and embedding. **g** As **e** but for a lateral newborn Asian elephant trunk FSC (scale bar = 100 μm). **h** Volume rendering of an Asian newborn elephant trunk microCT scan with single FSCs and their respective whiskers segmented on the trunk tip and a proximal trunk piece. FSCs are color-coded depending on their lengths. Left, view of the whole scan. Right, Volume rendering of the proximal trunk piece. **i** Follicle length in the lateral and ventral trunk and the trunk tip of an Asian newborn elephant (Hoa's Baby, see Table 1). One-way ANOVA, $P < 0.001$, post hoc Scheffé test for pairwise comparison (*$P < 0.05$, ***$P < 0.001$). Error bars indicate the standard deviation. RRC Rete ridge collar, MS Mesenchymal sheath, C Vibrissal capsule, S Vibrissal shaft, RoS Root sheath, VSS Vascular sinus spaces, DVN Deep vibrissal nerve.

**Table 1 Overview of the elephant specimen used for whisker count and FSC dissections.**

| Name | Species | Sex | Age (years) | Whisker count[*] |
|---|---|---|---|---|
| AM1 | *L. africana* | M | 0 (stillborn) | 773 |
| Indra | *L. africana* | F | 34 | 601 |
| Linda | *L. africana* | F | 34 | 658 |
| Unknown African | *L. africana* | ? | ? (adult) | 589** |
| Zimba | *L. africana* | F | 39 | 635 |
| Ali | *L. africana* | M | 23 | 468 |
| Burma | *E. maximus* | F | 51 | — |
| Dumba | *E. maximus* | F | 44 | 408 |
| Hoa's Baby | *E. maximus* | F | 6 days | 424 |
| Ilona | *E. maximus* | F | 45 | 292 |
| Naing Thein | *E. maximus* | M | 40 | 397 |
| Raj | *E. maximus* | M | 4 | 378 |
| Unknown Asian | *E. maximus* | ? | ? (adult) | 332 |
| Vilja | *E. maximus* | F | 61 | 335 |

[*]Refers to the number of whiskers on the trunk tip and the first eight skinfolds proximal to the tip.
**data partially extrapolated because the dorsal finger was missing.
Dissections of FSCs were done from Linda, Zimba, Burma and Hoa's Baby.

the trunk, whereas in Asian elephants no whiskers can be found in a small area posterior to the trunk tip. Moreover, in African elephants the trunk is more bulged in the area of the ventral whisker ridges, giving the whisker bands a more pronounced look.

On the dorsal trunk side, whiskers are distributed more evenly, with whisker density in the investigated newborn African elephant being highest on the tip and then gradually decreasing over the trunk lengths. In the newborn Asian, however, whisker density is high in the tip area and on the proximal trunk and lower in the middle part of the dorsal trunk (Fig. 7c, d).

Elephants commonly use the ventral side of the trunk to balance objects. Thus, ventral trunk whisker ridges are in contact with objects during these balancing behaviors (Fig. 7e). Photographing and filming such ventral-trunk-object contacts was extraordinarily difficult. The reason is that many of the ventral trunk movements were rather fast and that visualizing ventral trunk whiskers required close-up filming; the inset in Fig. 7e shows that whiskers indeed contacted balanced objects. We hypothesize that this contact between the whisker bands and

the object plays a crucial role in keeping balanced objects centered on the trunk. Interestingly, the area proximal to the trunk tip, where whiskers are missing in the Asian elephant, is the place where Asian elephants—according to our behavioral observations—tend to clamp smaller objects (Fig. 7f). The ventral ridge whisker arrays show a high whisker density. This high-density arrangement is also evident from a microCT scan of a ventral trunk piece from a newborn Asian elephant (black box in Fig. 7b), in which we segmented the corresponding follicles. Figure 7g shows volume renderings of the trunk piece with volume renderings of all segmented follicles depicted in the lower part of the figure. All ventral ridge follicles share a similar ventral/ forward orientation. In conclusion, we show that

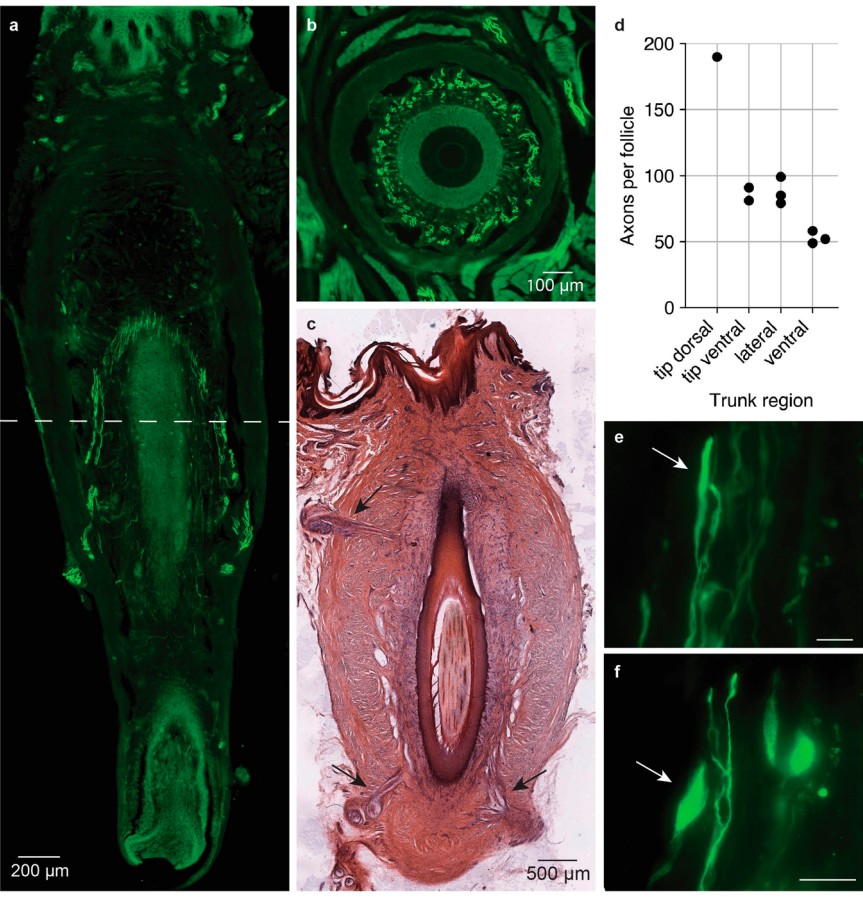

**Fig. 5 Innervation of elephant trunk follicle-sinus-complexes. a** Longitudinal section of a newborn Asian elephant trunk whisker FSC stained for Neurofilament H using immunohistochemistry. The dashed line indicates the level of the transversal section shown in **b**. Axon bundles enter the follicle in multiple nerves. Accordingly, we observed the axons penetrating the capsule in this and adjacent sections and ascending through the follicle while distributing evenly around the hair shaft. **b** Transversal section of a newborn Asian elephant trunk whisker FSC stained for Neurofilament H using immunohistochemistry. The approximate level of the section is indicated by the dashed line in **a**. **c** Longitudinal section of a hematoxylin-eosin stained adult Asian elephant trunk whisker FSC. Arrows indicate multiple nerves penetrating the follicle at different levels. In addition to their distinct appearance in hematoxylin-eosin staining, we identified nerves by Neurofilament H antibody labeling of alternating serial sections. **d** Number of axons per follicle counted for follicles of different trunk areas. The number indicated is a cumulative axon count corresponding to the sum of all axons counted in the various nerves innervating the respective follicle. Data refers to one newborn Asian elephant (Hoa's baby, see Table 1). **e** Micrograph of a lanceolate ending (indicated by an arrow) at the upper third level of a newborn Asian elephant trunk FSC stained for Neurofilament H using immunohistochemistry (scale bar = 20 μm). **f** Micrograph of lanceolate-like droplet-shaped nerve ending (indicated by an arrow) at the midlevel of a newborn Asian elephant trunk FSC, conventions as in **e**.

elephants have specialized ventral ridge whisker arrays that contact objects balanced on the ventral trunk.

## Discussion
Elephants have dense arrays of trunk whiskers that differ markedly between African and Asian elephants. Elephant trunk whisker length is use-dependent and lateralized probably as a result of lateralized trunk behavior. Whisker follicles are heavily innervated by (87 ± 40, mean ± sd) axons and the follicle lengths varies with trunk region. Elephants do not whisk. Ventral trunk ridge whisker arrays might contribute to object balancing. The elephant whisker system appears to be shaped by specialized trunk behaviors.

We find that elephants have numerous whiskers. African elephants have ~1.7x more trunk tip whiskers than Asian elephants (Fig. 1g). Compared to histological samples from lab-reared animals, our sample of elephant material has obvious limitations, as it consists of zoo animals that died of natural causes at varying ages. Still, we think the substantial number of elephants that entered our study leads to robust conclusions. Whiskers are

particularly dense on the trunk tip with the exception of the finger tip, where elephants pinch objects. Furthermore, there are high-density whisker arrays at both sides of the ventral trunk, which we discuss below. In conclusion, we are surprised that the marked difference in whisker number between African and Asian elephants has found so little attention so far.

Most, if not all mammals including newborn elephants have symmetrical facial whiskers. Adult elephants, however, are an exception, and in all adult elephant trunks, we investigated trunk whisker length is strongly lateralized (Fig. 2). We suggest that two factors give rise to this elephant peculiarity. The first one is the well-established lateralization of trunk behavior[8–11] that results in asymmetrical abrasion of the trunk whiskers. In line with this idea, we observed that Zoo elephants had trunk whisker abrasion patterns consistent with their behavioral lateralization (Supplementary Fig. 2). Secondly, elephants do not seem to replace whiskers by the dual whisker follicle mechanism observed in rodents[20]. In rodents, whiskers are replaced by a younger whisker growing in the same whisker follicle. The old whisker falls out when the young whisker reaches the length of the old whisker.

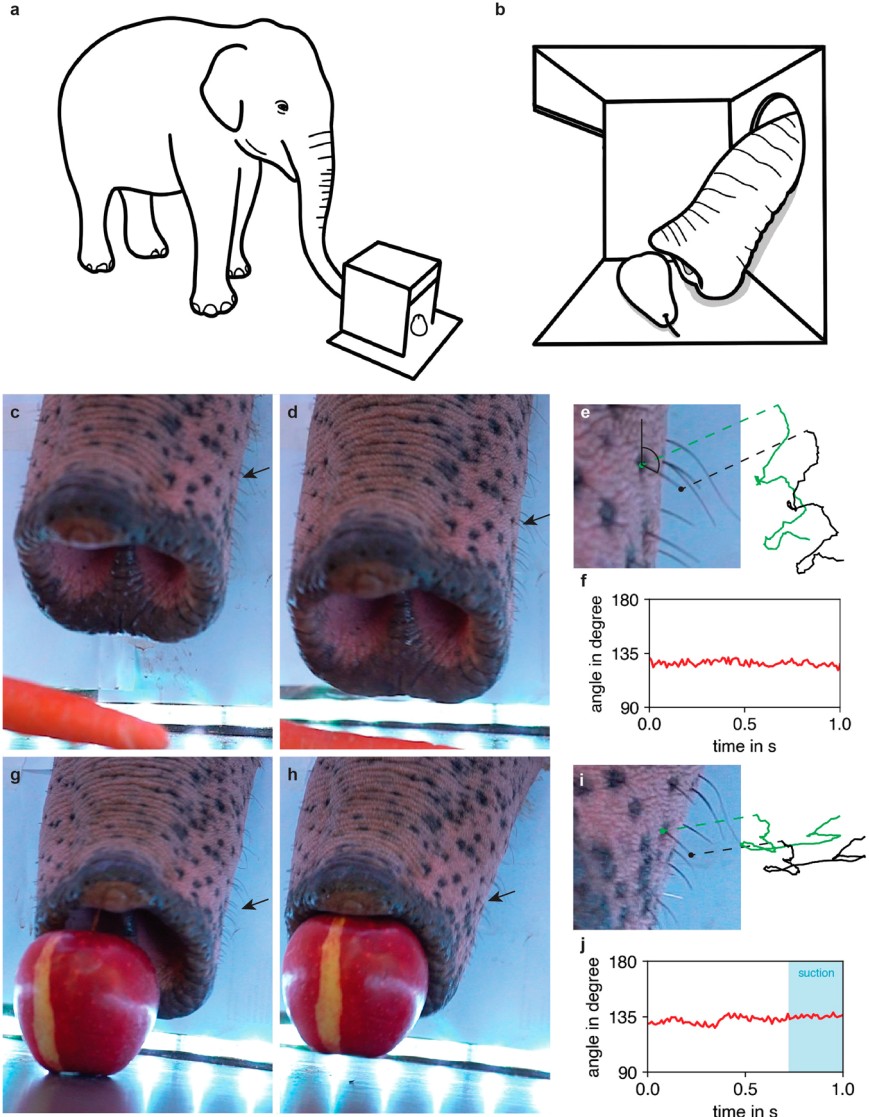

**Fig. 6 Elephants do not whisk (even during haptic pinching and vacuuming). a** Schematics of the experimental setup used for investigating whisker movement during haptically controlled retrieval of fruit from a wooden box. **b** View from the side of the box during the grasping task pictured in **a**. **c** Still of a grasping video clip, in which the female Asian elephant Anchali pinches a carrot; the frame shown is the first one we analyzed in the clip. Also see Supplementary Movie 1. The arrow points to the tracked whisker. **d** The final frame we analyzed in the same video clip. **e** Left, close-up view of the tracked whisker with the whisker base marked by a green dot and the tip of the whisker marked by a black dot. We measured the angle between the vertical of the trunk and the chosen whisker (marked in black). Right, trajectories of the base (in green) and tip (in black) during the tracking time of 1 s. The trajectories look identical indicating a lack of relative whisker movement (whisking). **f** Angle between the whisker and the vertical over the tracking time. Measurements were taken as indicated in **e**. Little or no whisker movement is observed during the pinching behavior. **g** Still of a video clip, in which Anchali vacuums an apple. Conventions as in **c**. Also see Supplementary Movie 1. **h** The final frame we analyzed in the same video clip. **i** As in **e**, but for the vacuuming behavior. **j** Angle between the whisker and the vertical over the tracking time. The angle is shown in **e**. The onset and time of suction is marked in blue and was inferred from the audio trace of the video. See also Supplementary Movie 1.

Therefore, the whisker array is length-wise always in 'perfect shape'. Elephants do not seem to have such a dual whisker replacement strategy and we wonder how exactly whisker growth unfolds in these animals. In any case, many whiskers—and in particular the trunk tip whiskers—are much shorter in adult elephants than in newborn elephants. Our understanding of whisker abrasion and whisker length in elephants leads us to question the presence of specialized vellus vibrissae in elephants, as described by Rasmussen and Munger (1996)[15]. These authors suggested—based on the histological investigation of a single trunk tip—that vellus vibrissae are specialized skin-internal whiskers on the elephant trunk finger. While we agree that trunk finger whiskers can be very short, our investigation of

multiple trunk tips across elephants of different age points more towards widely varying abrasion patterns than to specialized vellus vibrissae. Specifically, we found no evidence for the presence of vellus vibrissae in newborn elephants.

Similar to trunk whisker length, lateralization of tusk length has also been reported in African elephants[23]. They predominantly use one of their tusks as a 'master tusk' for tusk-involving behaviors, which leads to lateralization in lengths. The difference in tusk weight increases with the total weight of the tusks, and therefore with the age of the elephant[23]. This is comparable with our observation of same-length trunk whiskers in newborn elephants versus a strong lateralization of whisker lengths in adult elephants.

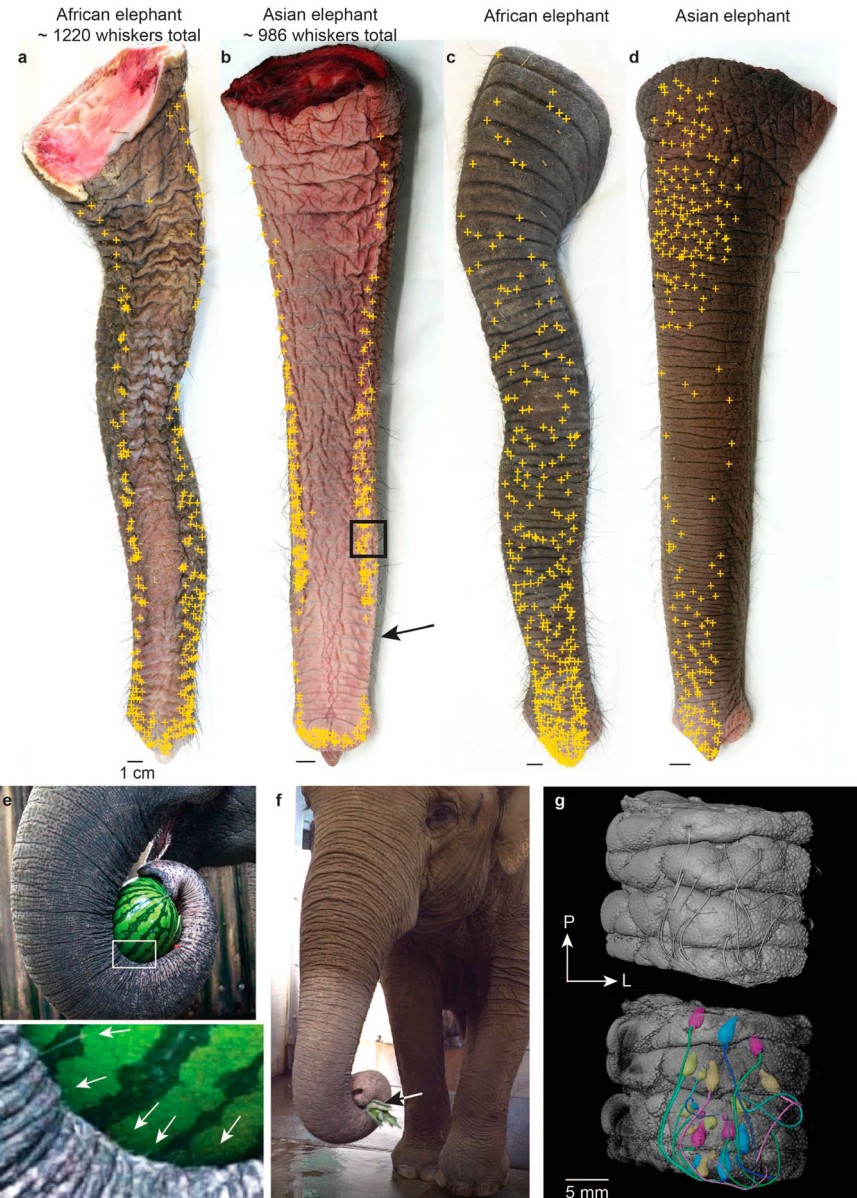

**Fig. 7 Ventral trunk ridge whisker arrays and their behavioral contact patterns. a** Photograph of the ventral side of a newborn African elephant trunk with whisker positions marked in yellow. All trunks investigated show two distinct whisker bands on the ventral trunk. In African elephants, whisker bands are more marked than in Asian elephants and extend over the whole trunk length. On top, we provide the total number of whiskers counted on the entire trunk of this newborn elephant. **b** Photograph of the ventral side of an Asian newborn elephant trunk with whisker positions marked in yellow. In Asian elephants, whiskers are missing in the 'clamp- zone' (black arrow). On top, we provide the total trunk whisker count. **c** Photograph of the dorsal side of a newborn African elephant trunk with whisker positions marked in yellow. **d** Photograph of the dorsal side of a newborn Asian elephant trunk with whisker positions marked in yellow. **e** Upper, photograph of an Asian elephant balancing a watermelon. Lower, high magnification view of the same picture, with white arrows pointing to ventral ridge whiskers that are visibly in contact with the melon during balancing. **f** Photograph of an Asian elephant clamping a pineapple; ventral ridge whiskers are distinctly missing in the trunk region posterior to the tip, where Asian elephants clamp objects. **g** Upper, volume rendering of a microCT scan of an iodine-stained trunk piece with ventral ridge whisker arrays from a newborn Asian elephant (black box in **b**). Lower, volume rendering of segmented whisker follicles. Note the ventral position and orientation of all whiskers. P posterior, L lateral.

In summary, we suggest the markedly lateralized trunk whiskers of adult elephants reflect the lateralized trunk behavior of these animals.

Elephants have large whisker follicles that differ strongly from rodent whisker follicles. Even though elephant whiskers are only slightly longer than rat whiskers, their whisker follicles are multiple times larger than rat follicles. This rather striking difference might relate to the difference in body size or could be related to the specialization of rodent whiskers for fast whisking movements. Elephant whisker follicles lack ring sinus and ring wulst, a prominent feature of the whisker follicles of many mammals[19,21,24]. These structures, in conjunction with specialized club-like nerve endings, are thought to allow a more fine-grained neural response and enable sensing of delicate movements of the whisker[25]. Hence, we reckon elephant whiskers might not be specialized for sensing very delicate deflections. Other species lacking a ring sinus include rhesus monkeys[26] and tammar wallabys[27].

Our findings indicate lanceolate nerve endings to dominate the sensory afferents in the elephant FSC (Fig. 5). Previous studies in

the rat reported isotropic neural responses of lanceolate nerve endings, i.e. direction invariant receptor responses upon tactile stimulation[28]. We further found that each elephant trunk FSC is innervated by multiple nerves. This observation has also been made for closely related species: The facial whisker of manatees are innervated by ~5 nerve bundles per FSC[17] and in hyraxes, both facial and postfacial whisker are innervated by two deep vibrissal nerves per FSC[16]. Species that have more than one deep vibrissal nerve also include rhesus monkeys[29] and bottlenose dolphins[30]. We counted an average of 87 innervating axons per follicle. If we multiply this number with the whisker number (493) per hemitrunk, we arrive at an estimate of ~42,900 whisker afferents per hemitrunk in Asian elephants. In a previous study[2] we counted about 400,000 axons in the Asian elephant infraorbital nerve; accordingly, we estimate that whisker afferents account for about 11% of the trunk afferents; since our counts did not include the (minor) superficial vibrissal nerve, we think of this estimate as a lower bound value. In comparison, in their 2001 study about the microanatomy of manatee FSCs, Reep et al. reported an estimate of 30,000 axons innervating the manatee oral disk whisker follicles, which are mainly used for tactile exploration[17]. In the same study, the authors also found, that the anatomy of the manatee's facial FSC varies with face region (bristle field). Specifically, the FSCs of different regions differ in lengths, widths and axon count, with a positive correlation between ring sinus parimeter/area and the number of innervating axons[17]. While we also found differences in lengths and axon count between FSCs of different trunk areas (Figs. 4 and 5), the two parameters don't seem to correlate in the elephant.

In summary, we find elephant trunk whisker follicles are innervated by multiple nerves comprised of numerous axons. The lack of ring sinus and ring wulst (whisker structures associated with fine-grain sensing and high-frequency discharges) might indicate that the role of elephant whiskers is primarily the sensing of coarse tactile stimuli.

Our study uncovers numerous systematic whisker differences between African and Asian elephants. These differences include whisker number, thickness and follicle shape. We wonder if the differences in the behavioral ecology of the elephant species, such as mixed food preferences of African elephants[31] compared to the preferential grazing/browsing observed in Asian elephants[32] resulted in more numerous and tougher whiskers of African elephants. We therefore wonder if the thicker whiskers of African elephants might be instrumental in feeding on shrubs. The cylindrical whisker morphology of the elephant differs strikingly from the tapered whiskers in the rat (Fig. 3). Mechanically, tapered whiskers are thought to play a pivotal role in actively sensing the environment through sweeping motions of the whisker, as this geometrical feature prevents getting stuck at objects[33]. This goes along our findings of no active whisking behavior in elephants (Fig. 6), suggesting selection pressure for tapered whiskers was absent in the evolution of elephants. In manatees, the maximal diameter of facial whisker ranges from 0.5–2 mm depending on bristle field area[34] and manatee whiskers are therefore on average thicker than elephant trunk whiskers, a difference that is likely related to the aquatic lifestyle of manatees[35].

It is evident from our high-resolution videography that elephants do not whisk. Importantly, we obtained this result in a behaviorally meaningful context, i.e., during haptically controlled grasping. Moreover, we could also show, that whisker movements are absent during haptically controlled vacuuming (Fig. 6, Supplementary Movie 1). This absence of whisker movements during vacuuming suggests that elephant whiskers show no respiration-associated whisker movements; whisker-movement-respiration coupling is common in mammals[22]. The lack of such coupling in

elephants enforces the idea that elephant whiskers are genuinely immobile. The rigid skin embedding of whiskers and the lack of capsular whisker musculature, as observed in our microCT scans, are in line with our observation that elephant whiskers do not whisk. We reckon that elephants do not require whisker-specific mobility, as their trunk mobility dramatically increased in evolution. In summary, we suggest that high mobility and flexibility of the elephant trunk might have made the autonomous whisker-mobility of trunk whiskers obsolete.

The ventral trunk ridge whisker arrays are an elephant whisker specialization. These ventral whiskers do not cover the entire trunk, but rather form two rows at the ventral ridges of the trunk. Particularly, in adult elephants, these whiskers are more conspicuous in African than in Asian elephants. In Asian elephants, these whiskers are missing directly behind the trunk tip, where Asian elephants often clamp objects. Our informal observations suggest that the ventral trunk ridge whiskers are involved in object balancing on the ventral trunk—a very common elephant behavior.

Trunk whiskers of elephants differ markedly from the facial whiskers of other mammals. In many small mammals, whiskers are thin, tapered, mobile, symmetrically arranged around the snout and function in peri-snout sensing. In contrast elephant trunk whiskers are thick, non-tapered, immobile, lateralized and are arranged in specific high-density arrays on the ventral trunk and the trunk tip. We suggest unique trunk whisker characteristics evolved to provide a haptically controlled action space for the extraordinary manipulative capacities of the elephant trunk.

## Methods

**Elephant and rat specimens**. All specimens used in this study came from zoo elephants and were collected by the IZW (Leibniz Institute for Zoo and Wildlife Research, Berlin) over the last three decades in agreement with CITES (Convention on International Trade in Endangered Species of Wild Fauna and Flora) regulations. Specimen reports and CITES documentation for all animals included are held at the IZW. All animals included in the study died of natural causes or were euthanized by experienced zoo veterinarians for humanitarian reasons, because of serious health complications. Table 1 gives an overview of the specimens of Asian elephants (Elephas maximus) and African elephants (Loxodonta africana), along with the age and trunk tip whisker counts derived from these animals.

We collected rat whiskers and FSCs post-mortem from 6-week-old male long-evans rats killed under a permit approved by the State Office for Health and Social Affairs committee (LAGeSo) in Berlin (Animal license number: G0095-21 / 1.2).

**Photography, whisker counts and whisker thickness measurements**. We photographed the trunk tips from all sides using a Sony α 7 R III camera with a Sony FE 90 Mm/2.8 Macro G OSS objective. The majority of the trunk tips photographed and analyzed were either fixed in 4% formaldehyde solution or were frozen at −20 °C; a minority of samples consisted of fresh post-mortem material. For counting the whiskers, we adjusted the color curves of the photographs in Adobe Photoshop (Adobe Systems Incorporated) for maximum visibility of the whiskers. The count was performed manually using the multi-point tool in ImageJ (Rasband, W.S., ImageJ, U. S. National Institutes of Health, Bethesda, Maryland, USA). For every specimen, we counted the whiskers on the tip and on approximately eight segments (skinfolds) proximal to the trunk tip due to the limited availability of whole trunk samples. The first author (ND) obtained whisker counts from six African elephants and seven Asian elephants using five pictures of each sample taken from different angles. For two trunk tips a coauthor (BG) recounted whisker numbers and we obtained similar numbers (a -5%, and +11% deviation); all counts reported are the first author counts. Details on species and age of each specimen are provided in Table 1. For one African elephant, the whisker count was extrapolated due to an incomplete sample. Additionally, we derived whole trunk whisker counts from one newborn Asian (Hoa's Baby) and one newborn African (AM1) elephant. For two adults from each species (n = 4; Ilona, Unknown Asian, Linda and Zimba, see Table 1), we measured whisker thickness using ImageJ from the same pictures that were taken for the whisker count. The measurements were taken on the widest point of each whisker, just where it protrudes from the surface of the skin. For each specimen, we measured the thicknesses of ten whiskers on each of four different areas: The trunk tip, the dorsal side of the trunk, the ventral side of the trunk, and both sides of the trunk. Our observations on whisker length and abrasion were made from pictures of the trunk samples (n = 6 African elephants and n = 9 Asian elephants) and n = 3 Asian zoo elephants during routine handling (see below).

**Experimental procedures with zoo elephants**. Behavioral experiments with Asian elephants were conducted in the Berlin Zoological Garden. Our experimental procedures were evaluated by the regional government, which ruled that a formal animal experimentation permit is not required given the non-invasive nature of our procedures (LAGeSo StN-statement 19.07.2021). We observed several Asian elephants (n = 5) during routine handling and haptically controlled grasping in the 10-year-old female Asian elephant Anchali. Anchali was trained to retrieve fruit from a closed box with a hole for the trunk (see Fig. 6), a behavioral setting that required to localize and grasp or vacuum various fruits (carrots, pears, bananas, apples) under haptic and olfactory control. We also obtained ad hoc information about the elephant habits (left- vs right-trunker) from the animal caretakers.

**Videography and whisker tracking**. Videography of elephant whiskers was challenging because whiskers are thin, elephants move their trunks very fast and we needed to keep a distance from the animals for security reasons (even though the elephant we worked with were well-habituated to humans). The best footage of trunk whiskers and the trunk tip was obtained with the elephant Anchali in the box-fruit-retrieving task described above. Under other circumstances resolving trunk whiskers was difficult or impossible and such difficulties also limited our study of ventral trunk whiskers.

We obtained videos of the behavioral experiments described using a Sony α 7 R III camera with a Sony FE 16-35 mm F2.8 GM E-Mount objective. The frame rate was set to 100 Hz. For tracking of single whiskers, we imported video clips of single events of fruit retrieval (n = 3) from the box in ImageJ and analyzed them manually by following base and tip of single whiskers (n = 5, from both sides of the trunk) over an interval of 1 s using the multi-count tool. We exported base and tip coordinates and plotted trajectories and the angle between the whisker and the vertical using Python's Matplotlib package. We calculated the angle using the following formula with Δx and Δy being the difference between the base and tip x and y cartesian coordinate for each timepoint, respectively.

$$\alpha = 90° - \sin^{-1}\left(\frac{\Delta y}{\sqrt{(\Delta x)^2 + (\Delta y)^2}}\right) \quad (1)$$

We identified the onset of suction for apple vacuuming using the audio track of the video.

**Fixation, sectioning and hematoxylin-eosin staining**. Whole trunks or trunk tips were fixed in 4% formaldehyde for several months. Some of the samples were frozen and thawed prior to fixation. We either dissected Follicle-Sinus-Complexes (FSCs) from the surrounding tissue or cut out small cubes of tissue containing one or multiple FSCs. For cryosectioning, the tissue was transferred into a 30% sucrose solution in phosphate buffer and left for at least 24 h prior to sectioning for cryoprotection. We then embedded the tissue in tissue freezing medium (Leica Biosystems, Catalog Nr. 14020108926) and cut it into 40 μm sections either perpendicular or parallel to the skin surface using a freezing microtome. For hematoxylin-eosin staining sections were mounted and dried for at least 24 h, directly followed by staining with hematoxylin-eosin solution. Pictures of the hematoxylin-eosin stainings were acquired with a MBFCX9000 camera (MBF Bioscience, Williston, USA) on an Olympus BX51 microscope (Olympus, Japan) using Neurolucida (MBF Bioscience, Williston, ND) software. We described characteristic structures and overall anatomy of the FSC using hematoxylin-eosin stained sections from n = 8 follicles (from Hoa's Baby, Burma, Unknown Asian and Zimba, see Table 1).

**Immunohistochemistry/ antibody characterization**. We incubated the sections in a blocker of 0.1 M PBS, pH 7.2, with 0.75%Triton X-100 and 2,5% Bovine Serum Albumin (BSA) for an hour at room temperature before incubating them with a polyclonal antibody against Neurofilament-Heavy (NF-H) (1:1000, Millipore, Catalog Nr. AB5539) in 0.1 M PBS, pH 7.2, with 0.3%Triton X-100 and 1% BSA for 48 h at 4 °C. We then washed and incubated the sections in a blocking solution containing 1% BSA in 0.1 M PBS and Goat anti-chicken IgY secondary antibody conjugated to Alexa Fluor 488 (1:1000, Invitrogen, Catalog Nr. A-11039) at 4 °C overnight. The next day we washed, mounted and coverslipped the sections using a mounting medium (Fluoromount G, SouthernBiotech, Catalog Nr. 0100-01). We took micrographs of the slides using a Leica DM5500B epifluorescence microscope (Wetzlar, Germany). We stained sections of n = 17 follicles from three different elephant samples (Hoa's Baby, Burma and Zimba, see Table 1) using immunohistochemistry. To gain a better understanding of the relative importance of the trunk whiskers as sensory structures we counted the axon numbers per FSC (n = 9, FSCs from Hoa's Baby) using serial cross-sections in ImageJ.

**Iodine staining and micro-computed tomography (microCT) scanning**. We took all samples used for microCT scanning from trunk samples that were fixed in 4% formaldehyde for several months. To characterize follicles from different trunk regions, we cut a newborn Asian elephant trunk (Hoa's Baby, see Table 1) in half sagittal and stained it in 1% iodine solution for 33 days to enhance tissue contrast. To characterize ventral FSCs, we cut out a small piece (2 cm × 1.5 cm × 1 cm) of the same baby Asian elephant trunk and stained it in 1% iodine solution, followed by

2% iodine solution for 4 days each. We also dissected single FSCs (n = 4, from Burma, Zimba and Linda, see Table 1) from trunk pieces of different trunk tip samples and stained them in 1% iodine solution for 48 h before scanning. We scanned single FSCs by embedding them in a mixture of 2.5% gelatin and 1% agarose to prevent the sample from moving in the scanner while minimizing iodine bleeding. For comparison of structure and size, we dissected the follicle of a rat δ-whisker follicle from a facial whisker pad, that was fixed in 4% formaldehyde for >20 days. The follicle was stained in 1% iodine solution for 24 h.

For comparison of whisker thickness and shape, we pulled lateral trunk whiskers of one adult Asian and African elephant each (Burma and Zimba, see Table 1) and a rat δ- whisker out of the follicle and scanned them embedded in 2% Agarose after staining them in 1% iodine solution for 10 min. All iodine solutions were prepared by dilution of 5% Lugol's iodine in distilled water. We performed the scans using a YXLON FF20 CT scanner (YXLON International GmbH, Hamburg, Germany) of our institute. The scans of the newborn elephant trunk were performed at 115–120 kV source voltage, 20–25 μA source current, and 1000 ms exposure. The scans of the FSCs were performed at 36–55 kV, 81–120 μA, and 2000 ms exposure. Single whiskers were scanned at 50 kV, 81 μA, and 333 ms.

**Statistics and reproducibility**. If not otherwise noted all errors refer to the standard deviation.

We tested for normality using Shapiro-Wilks tests. Levene's tests were used to test for equal variances between populations. If the populations had equal variances, we tested the difference between the population means using a student t-test, otherwise, we used Welch's t-test. All t-tests were two-sided, and the null hypothesis was rejected when p < 0.05. We tested the significance of differences in the mean of more than two populations using a one-way ANOVA. Post-hoc testing for pairwise comparison of groups was done using a Scheffé test if the populations showed equal variances, otherwise a Tukey test was used for post-hoc testing. We determined differences in the laterality of whisker length between newborn and adult elephants using Fisher's exact test, with data from both elephant species pooled.

The statistical analysis was done using Pythons Numpy, Scipy and Scikit-posthocs packages. We used Numpy to calculate the means and standard deviations, Scipy for t-tests, the ANOVA, Shapiro- Wilks tests and Levene's tests and functions from the Scikit-posthoc package for post-hoc testing.

We clarified the sample sizes of the conducted experiments in the respective method sections.

**Segmentation and CT scan analysis**. For the segmentation of the trunk whisker FSCs we used an extended version of the Amira software (AmiraZIBEdition 2021, Zuse Institute). For that, we marked the area of the follicle on every tenth slide and the volume in between two marked areas was extrapolated. We obtained the lengths of the follicles using the Length3D function of Amira's label analysis module. In total, 43 FSCs of Hoa's Baby were segmented, n = 9 in the tip region, n = 8 lateral FSCs and n = 26 ventral FSCs.

**Drawings**. Outlines in Figs. 1e, f and 6b were drawn by ND using reference images taken by the authors (ND and LVK). Figure 6a was drawn by LVK from video footage of the experiment. The outlines in Fig. 2f were drawn by ND using Fig. 1a, b of Maier & Brecht (2018)[20] as a reference. The dot plots and density maps in Figs. 1e, f and 2f were realized using Python's Matplotlib library.

**Reporting summary**. Further information on research design is available in the Nature Portfolio Reporting Summary linked to this article.

## Data availability

All data needed to evaluate the conclusions in the paper are present in the paper and/or the Supplementary Materials. Additional data reported in this paper are shared on a publicly accessible repository (https://gin.g-node.org/elephant/Deiringer). This paper does not report original code.

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

## Acknowledgements

We thank the Berlin Zoological Garden and in particular Rouven Schulze, Konstantin Becker, Lucas Baum, and Alina Kobin. We also thank Katriona Guthrie-Honea, Luke Longren, Andreea Neukirchner, Maik Kunert and Tanja Wölk. Several zoological institutions contributed, in particular the Berlin Zoo (Germany) for behavioral observations, and for anatomy Zoo Augsburg (Germany), Opel-Zoo Kronberg (Germany), Zoo Poznań (Poland), Tierpark Hagenbeck (Germany), and the Elefantenhof Platschow (Germany). Supported by BCCN Berlin, Humboldt-Universität zu Berlin and the Deutsche Forschungsgemeinschaft (DFG, German Research Foundation) under Germany´s Excellence Strategy—EXC−2049—390688087.

## Author contributions

Conceptualization, N.D., U.S., T.H. and M.B.; Methodology and materials, N.D., U.S., L.E., L.V.K, C.S., B.G., S.H., G.F., F.G., R.B., A.O., T.H. and M.B.; investigation, N.D., U.S., L.E., L.V.K, C.S., B.G., R.B., A.O., T.H., and M.B. Formal analysis, N.D., L.E., and M.B.; writing N.D., T.H. and M.B.; supervision, M.B.; funding acquisition, M.B.

## Funding

## Competing interests

The authors declare no competing interests.
