## [Peer Review File · Communications Biology]

Reviewers' comments:

Reviewer #1 (Remarks to the Author):

A lovely anatomical and behavioural study of elephant whiskers. A systematic approach to the work, leading to some lovely findings and beautiful figures. To my knowledge, all of these findings are entirely novel. However, the work is very focussed (only on elephants really), and needs to be put in the wider context to appeal to a broad readership. I suggest adding some sentences to the abstract, introduction and discussion to do this. Please also see my specific comments below.

ABSTRACT

1. good, but focusses a lot on results. A greater justification of looking at whiskers, and the wider implications of the work would be appreciated

INTRO

2. "It is therefore not surprising that the trunk receives massive tactile innervation, weighing about 1.5 kg in elephant cows" Flipping from innervation to weight is not clear

3. Temper your language where you can I.e. highly, massive etc

4. Very focussed literature, can you also justify the wider implications of this work?

METHODS

5. Can your imaging capture the tip? Comment on accuracy/error of imaging and measuring here

6. Need to clearly state sample numbers in text (I.e. animals and whiskers) for immunohistochemistry section

7 Need to justify only having one rat whisker, and why you have selected that particular whisker

RESULTS

8. Switching from past to present tense, needs checking throughout

DISCUSSION

9. Could lateralisation of whisker length be caused by wear associated with lateralisation of trunk-use? Please comment on that.

10. As well as lateralization of length, how about density and innervation?

11. Nice parallels with hyrax and manatee. I think more than one Deep vibrissal nerves has also been observed in some species of cetacean – could look wider than just this group?

12. Difference in thickness of manatee and elephant whiskers is likely due to manatees being aquatic (see Dougill's paper)

13. Can you finish on a stronger conclusion to help put your findings in the context of the general mammalian whisker sensing literature perhaps?

FIGURES

14. Figures were not properly shown for me, as they were imported into a word document and were overhanging the page, so could not view the right hand panels of the figure. This is not at all helpful!

15. Clearly state sample numbers in Fig 1e-g

16. Figure 3a – needs a better image of rat whisker, you cannot really see it

17. Some beautiful figures here

SUPP VIDEO

18. Supp video: perhaps say trunk is haptically controlled with the whiskers, rather than whiskers? You can't really see the whiskers interacting with the object in the video, and it seems that it is the trunk rather, so the text here is misleading.

Reviewer #2 (Remarks to the Author):

Dieringer and colleagues have prepared and conducted a comprehensive study to describe whisker morphology, distribution, density, and innervation in two species of elephants. The study is novel in that it examines the whiskers of the trunk, rather than the skin on which previous literature has focused. The data are technically sound; however, it was difficult in some areas to assess the strength of the evidence for the conclusions since I had trouble following the sample sizes for the different study components, and the information provided on statistics in the methods was scarce. I recommend that the authors be far more explicit in the text (and potentially a table) with the sample sizes and whether their results apply to all ages, just adults, or just the calves. The study suggests that for both species, (1) whisker distribution and density are variable across the trunk with lateralization that appears to correlate with use, (2) the whisker morphology is cylindrical and not tapered, (3) whisker whisking is not a behavior supported by observations nor by whisker FSC morphology, and (4) whisker FSCs are highly innervated. Whisker density at the trunk tip seems to be higher in one species and the species show slightly different follicle morphology, although the authors could more clearly lay out how this could reflect differences in behavior or ecology. I appreciate the authors' effort to maximize useable data from collaboration with zoological facilities, which provide valuable resources for collaborative research.

I have provided detailed comments to the manuscript as a separate pdf document, and I have included an edited Word document with line numbers added to aid in my comments. I recommend that the manuscript be condensed in a few sections and improved in clarity (specifics indicated below), and the Discussion especially could be strengthened. I've also included some grammatical and syntax suggestions, but I recognize that these reflect personal writing style.

Reviewer #3 (Remarks to the Author):

The paper by Deiringer et al. reports characteristics of whiskers of elephant trunks. They offer fundamental information about trunk whiskers with valid methodology, and I was surprised that they had not been studied before this study (this was well reflected in the simple, concise title of this study). Therefore, I believe this study is important to better understand not only elephant behavior but also inter-species diversity of how whisker anatomy shapes mammalian behavior. They set clear questions to be addressed in this study, which makes readers easily understand their points. The strength of this study is that they well-integrate results from both anatomical and behavioral aspects. For example, the asymmetry of the whisker trunk length is noteworthy. They give important insights that trunk whisker does not whisk, unlike rats. Their argument that elephants have dramatically improved trunk mobility over the course of evolution and therefore do not need specific mobility of whiskers is very interesting. They also give a next question about how whisker grows in elephants, which would facilitate further research to follow. Overall, their findings about elephant whiskers, which vary according to region, age, species, and behavior will give us fundamental new insights to better understand elephants. Whisker studies in mammals are highly biased in rodents, and therefore this study has an important piece to fill the fundamental gap. I only have some minor comments.

Specific comments

1. Line 269: The statement here seems different from that in the introduction, where you state that "while African elephants tend to pinch objects with their two fingers, Asian elephants tend to grasp/wrap objects with their trunk. Please clarify this point.
2. Line 340: Figure 4i. Is this data from African or Asian elephants? Please clarify this in the figure legend. This comment is applied to Figure 5d.

Reviewer #4 (Remarks to the Author):

Please see the attached annotated MS word document of the manuscript.

Thank you for allowing me to review "Elephant Trunk Whiskers." I find the topic and scope to be excellent and appropriate for Communications Biology.

The data provided in this manuscript is excellent and the ecological interpretations are clear and supported (mostly*).

Throughout the manuscript review attached, I have provided suggestions on how to improve the clarity and flow of the document. Small additions like "define what a vibrissa is and what its key features (i.e., sinus?) are" may help assist future readers to navigate this interesting paper.

Overall, I found this manuscript to be very enjoyable, and left me wanting to know more about the histomorphology of the elephant trunk. This spark of curiosity is a direct result of the author's ability to show patterns between form and function.

I did have some questions about how the behavioral data and the "rat" data were included in this paper – as they were not outlined in the methods. Not included in my "comments" in the MS WORD document provided – was a question about body size and scaling. I wonder if you have found any size-related variation? Interestingly, I have found some size-related differences in vibrissa density across mammals and I have seen age-related differences in mechanoreceptor density as well. I would

have liked to have seen a small comment about the size of these two elephant populations. I fully appreciate that there are negligible differences in size between African and Asian elephants, but when you include stillborn, "baby," subadult, and adult specimens in a study – SIZE is a question that immediately comes to mind.

The discussion was amazingly clear and straightforward. However, a small concluding statement about the observed variation might be informative (directions for future work, e.g., histomorphology of the non-hairy (glabrous?) ventral surface of the trunk). I think that adding a concluding statement* may be helpful for readers like myself who wanted to know more, but assumed the authors did not collect the needed data to address my questions. However, the authors did present several hypotheses throughout – that could be "re-explored" or "re-stated" in a concluding statement.

In the end, I am very impressed with the author's holistic approach to comparative functional anatomy and I very much enjoyed reading it.

Reviewer #1:

A lovely anatomical and behavioural study of elephant whiskers. A systematic approach to the work, leading to some lovely findings and beautiful figures. To my knowledge, all of these findings are entirely novel. However, the work is very focussed (only on elephants really), and needs to be put in the wider context to appeal to a broad readership. I suggest adding some sentences to the abstract, introduction and discussion to do this. Please also see my specific comments below.

Comment: We are thankful for the overall positive assessment.

Change: We implement the suggested changes point-by-point.

ABSTRACT

1. good, but focusses a lot on results. A greater justification of looking at whiskers, and the wider implications of the work would be appreciated

Comment: The referee is right the abstract was very packed with data.

Change: We removed results and made the abstract more approachable.

INTRO

2. "It is therefore not surprising that the trunk receives massive tactile innervation, weighing about 1.5 kg in elephant cows" Flipping from innervation to weight is not clear.

Comment: We agree.

Change: We rewrote this sentence.

3. Temper your language where you can I.e. highly, massive etc

Comment & Change: We considered this point and removed hyperbole expressions.

4. Very focussed literature, can you also justify the wider implications of this work?

Comment & Change: As already pointed out above, we revised the ms and particularly the abstract to address the broader functional implications of our analysis.

METHODS

5. Can your imaging capture the tip? Comment on accuracy/error of imaging and measuring here

Comment: The referee raises an important question. It is indeed difficult to photograph or videograph the elephant trunk with sufficient resolution to resolve whiskers and whisker tips. Our efforts to visualize whiskers met with relatively good success in the elephant-cow Anchali, who is habituated to grasp fruit in a box. This is data shown in Figure 6 and our supplementary video. In this particular behavioral setting, we have a lot of footage and we are in the position to comment on whisker contacts. In other – less staged – settings, it was much more difficult to capture trunks with sufficient resolution to resolve whiskers. This is why our observations on ventral trunk whiskers (Figure 7) were only of limited photographic quality. The reasons are that whiskers are small, elephants move their trunks very fast, we

limited the use of flashlights, and need to keep a distance from the animals for security reasons (even though the elephant we worked with are well habituated to humans).

Change: In the revised ms we comment in more detail on the difficulties associated with whisker and trunk tip filming in the method section of the revised ms.

6. Need to clearly state sample numbers in text (I.e. animals and whiskers) for immunohistochemistry section

Comment & Change: We added ns throughout the ms.

7 Need to justify only having one rat whisker, and why you have selected that particular whisker

Comment: For the comparison we picked a large rat whisker, because the length was more similar to elephant whiskers.

Change: We explain the logic of our comparison choice in the revised ms. (Page 10, 2nd para).

RESULTS

8. Switching from past to present tense, needs checking throughout

Comment & Change: We rectified these inconsistencies.

DISCUSSION

9. Could lateralisation of whisker length be caused by wear associated with lateralisation of trunk-use? Please comment on that.

Comment: Yes, this is, what we think is happening and actually also meant to express in our paper. In the revised ms we devote more text and explanation space to communicate this concept better. Specifically, we point out how lateralized trunk behavior ('left- or right-trunker') in elephants is expected to lead to lateralized whisker abrasion.

Change: We clarified that we think that lateralization of whisker length is caused by wear associated with lateralized trunk use. (Page 9, 2nd para).

10. As well as lateralization of length, how about density and innervation?

Comment: We suspect that innervation patterns are determined prenatally. Since baby elephants are clearly not lateralized, we doubt there will be lateralized innervation. Also, whisker positions appear symmetric. While trunk use is heavily lateralized much like human hand use, there appears to be no genetic bias for certain lateralization (like the human left-hemispheric speech specialization and elephants come out 50:50 as left- and right-trunkers.)

Change: We did not alter the ms here, because our insights on the issue at hand are limited.

11. Nice parallels with hyrax and manatee. I think more than one Deep vibrissal nerves has also been observed in some species of cetacean – could look wider than just this group?

Comment: A great suggestion.

We looked at the cetacean literature and found that there is more than one nerve in bottlenose dolphins, but only one in the bowhead whale and harbor seal, according to

Mynett, N., Mossman, H. L., Huettner, T., & Grant, R. A. (2022). Diversity of vibrissal follicle anatomy in cetaceans. *The Anatomical Record*, 305(3), 609-621.

And

Gerussi, T., Graic, J. M., De Vreese, S., Grandis, A., Tagliavia, C., De Silva, M., ... & Cozzi, B. (2021). The follicle-sinus complex of the bottlenose dolphin (*Tursiops truncatus*). Functional anatomy and possible evolutionary significance of its somato-sensory innervation. *Journal of Anatomy*, 238(4), 942-955.

Change: We added this information and included the paper of Gerussi et al. We further added information about another species with more than one deep vibrissal nerve, the rhesus monkey.

12. Difference in thickness of manatee and elephant whiskers is likely due to manatees being aquatic (see Dougill's paper)

Comment: We agree and also note that the referee points out an excellent reference that we had missed.

Change: We cite and discuss the relationship between aquatic life-style and whisker thickness as observed Dougill et al 2020. (Page 16, second para).

13. Can you finish on a stronger conclusion to help put your findings in the context of the general mammalian whisker sensing literature perhaps?

Comment: Yes.

Change: We rewrote our conclusion para. (Page 17).

FIGURES

14. Figures were not properly shown for me, as they were imported into a word document and were overhanging the page, so could not view the right-hand panels of the figure. This is not at all helpful!

Comment: We are very sorry about this problem and we made sure to fix it.

Change: We fixed this problem.

15. Clearly state sample numbers in Fig 1e-g

Comment & Change: we added this information.

16. Figure 3a – needs a better image of rat whisker, you cannot really see it

Comment: We agree. We wanted to keep a picture where the three whiskers are photographed together to highlight the difference in shape and thickness.

Change: We enlarged the picture and exchanged it for the unedited version, to make the rat whisker as visible as possible next to the elephant whiskers.

17. Some beautiful figures here

Comment: Thanks.

Change: None.

SUPP VIDEO

18. Supp video: perhaps say trunk is haptically controlled with the whiskers, rather than whiskers? You can't really see the whiskers interacting with the object in the video, and it seems that it is the trunk rather, so the text here is misleading.

Comment: We think the referee's criticism of our task description was justified.

Change: We added information to the description of the behavioral task to clarify that we do not know to what extent whiskers are involved in the task at hand.

Reviewer #2 (Remarks to the Author):

Dieringer and colleagues have prepared and conducted a comprehensive study to describe whisker morphology, distribution, density, and innervation in two species of elephants. The study is novel in that it examines the whiskers of the trunk, rather than the skin on which previous literature has focused. The data are technically sound; however, it was difficult in some areas to assess the strength of the evidence for the conclusions since I had trouble following the sample sizes for the different study components, and the information provided on statistics in the methods was scarce. I recommend that the authors be far more explicit in the text (and potentially a table) with the sample sizes and whether their results apply to all ages, just adults, or just the calves. The study suggests that for both species, (1) whisker distribution and density are variable across the trunk with lateralization that appears to correlate with use, (2) the whisker morphology is cylindrical and not tapered, (3) whisker whisking is not a behavior supported by observations nor by whisker FSC morphology, and (4) whisker FSCs are highly innervated. Whisker density at the trunk tip seems to be higher in one species and the species show slightly different follicle morphology, although the authors could more clearly lay out how this could reflect differences in behavior or ecology. I appreciate the authors' effort to maximize useable data from collaboration with zoological facilities, which provide valuable resources for collaborative research.

Comment: The referee gives an overall positive assessment of our study and provides a concise summary of our findings. We are thankful for this positive assessment. The referee also criticizes that the sample size underlying our observations is not always clear.

Change: We added info on sample sizes and statistics throughout the ms.

I have provided detailed comments to the manuscript as a separate pdf document, and I have included an edited Word document with line numbers added to aid in my comments. I recommend that the manuscript be condensed in a few sections and improved in clarity (specifics indicated below), and the Discussion especially could be strengthened. I've also included some grammatical and syntax suggestions, but I recognize that these reflect personal writing style.

Comment: It is obvious to us that the referee put a huge amount of work into our ms and we found almost all criticisms justified.

Change: We implemented the referee suggestions point-by-point as indicated below.

Reviewer #2 additional comments**1. Principal strengths of the manuscript:**

- *The authors have combined multiple data streams (behavioral, immunohistochemical, morphological) to comprehensively describe elephant whiskers. I truly appreciate the time and effort it takes to combine these techniques that attempts to link structure and function.*

Comment: The referee appreciates our combined methodology.

Change: None.

- *The authors must be commended for creating such beautiful figures! They were a pleasure to look at and informative.*
- *I appreciate that the authors provide data in a public repository – a great practice!*

Comment: The referee comments positively on our Figures and the repository.

Change: None.

2. Principal uncorrectable weaknesses of the manuscript:

The authors obtained a range of ages and sexes for this study, but it seemed to me that sampling for some data streams may be bit biased – for example, FSC length seemed to be only sampled in calves. This is often the nature of this type of research with something other than a lab-reared model organism, and while it doesn't reflect poorly on the quality of the existing data, I have suggested places where the authors could be more explicit about the limitations for their conclusions. I did have difficulty tracking sample sizes for each arm of the study (which is something I've listed as a suggestion to improve below), so it's possible the sampling isn't as biased as it seems.

Comment: The referee notes that there are uncorrectable weaknesses in the underlying samples and that our material is not as controlled and complete as histological material obtained in an elective fashion from lab-reared animals. We agree the referee has a point concerning the limitations of our material. Having said that the referee's point is well taken, we would like to add that the material entering our study (histological material from ≥ 10 elephants) is substantial in comparison to other work in the elephant field. For example, the most significant study on the somatosensory histology of elephant trunk tip (Rasmussen & Munger 1996) is an $n = 1$ study (with respect to histological samples).

Change: We added info on sample sizes throughout the ms. We pointed out the limitations of our material in the discussion (page 14 2nd para).

3. Principal correctable weaknesses of the manuscript:

- *I recommend the authors develop a stronger title that provides information about either the methods used or the results/conclusions*

Comment: We debated various alternative titles and initially had also favored more complicated titles. In the end, however, the authors favored the current title and we'd prefer not to change it.

Change: None.

- *I suggest the authors provide some broader context statements about the study system and their findings in Introduction and Abstract. Right now that context doesn't come through as clearly as it could. Why are elephants interesting to examine and why might the whiskers be interesting? I think the authors could emphasize more the uniqueness of the trunk as a way to explore if patterns we see in other tactile whisker specialists is maintained in this specialized structure, as well as the importance of contributing data from non-model, non-lab-reared organisms to the literature.*

Comment: The referee criticizes our abstract, a comment that we take seriously, specifically because it aligns with similar criticisms of referee 1.

Change: We entirely rewrote the abstract. We removed data from our summary and made an attempt to provide more context.

More details should be provided in the Methods, especially for (1) Statistics section, which only provides information on the language and packages used rather than the tests, and (2) numbers and ages of elephants used for each part of the study. The authors compile many different data streams, which is a strength of the manuscript, but

it is not always clear what the sample sizes and developmental stage are for each data type, which makes it difficult to assess the scope of the conclusions.

Comment: As already pointed out above we agree with this criticism.

Change: We added info on sample sizes and statistics throughout the ms.

- *In many sections, the Results include text that should be described in the Methods or the Discussion. I've pointed these out in my line-by-line comments.*

Comment: We agree with many of these suggestions.

Change: Please see our point-by-point response below.

- *I think the Discussion could benefit from a revision. Many sections feel like they are missing summarizing points for paragraphs -- literature has been added but it's not always clear how the points provide strong support or clarification for this study's findings. I'm not suggesting that they don't provide support, but rather the authors may find it helpful to work through the Discussion again to make sure that the conclusions of the study shine through. In other sections, not enough literature has been provided.*

Comment: The referee suggests a revision of our discussion. We agree with this sentiment.

Changes: 1. We rewrote the discussion. 2. Specifically, we made an effort to structure the different divisions of the discussion more clearly. 3. We completely overhauled our conclusion paragraph. (Pages 14-17).

- *For example, the section "Species whisker differences and comparison to previous work" - the authors end with a comparison to manatees, but do not provide any concluding statement to provide context to this comparison. It's not clear why manatees are the comparison or why the difference in thickness is informative. Given that manatees have a vastly different habitat and have highly differentiated facial and bodily vibrissal regions, it's not surprising that their whiskers are different than elephants.*

Comment & change: We rewrote the sections of the ms referring to manatees.

- *For example, "Trunk movements rather than whisker movements shape whisker contacts" is strong by directly linking the authors results to what's known for other mammals. However, the authors mention the evolution of trunk mobility in the last line without providing a citation or expanding further on how this would influence the mobility of elephant whiskers. I think the authors make a compelling point – that a highly dexterous/mobile trunk would reduce the selection pressure for a highly mobile whisker – but the point could be strengthened by making this statement a bit more direct.*

Comment: We agree.

Change: We implemented this suggestion and include the referee's reasoning on the interplay between selection pressures on trunk motility and whisker motility. (Page 16)

- One citation may be missing – I recommend the authors consider including it if helpful to the discussion.
- Sprinz, R. "The innervation of the trunk of the Indian elephant." *Proceedings of the Zoological Society of London*. Vol. 122. No. 3. Oxford, UK: Blackwell Publishing Ltd, 1952.

Comment: We consider the recommended literature to be highly relevant for our article and included it in the discussion of the literature.

Change: We included this reference.

4. Any sections that could be reduced or condensed?

- *I suggest being consistent with use of whiskers or vibrissae throughout manuscript - either is fine but the use of both may be confusing.*

Comment: We agree.

Change: We now use whisker throughout the manuscript consistently wherever it doesn't interfere with the existing terminology from the literature.

- *I recommend the authors adjust verb tense and change tense from passive to active whenever possible to make sentences more concise. I've provided examples below, but revising instances of these throughout the manuscript will improve conciseness.*
 - *"Elephants are almost constantly engaging their trunk" can be simplified to "Elephants constantly engage their trunks"*
 - *"Scanning of single FSCs was done by embedding them in a ..." can be simplified to "We scanned single FSCs by embedding them..."*

Comment & change: The referee is right, we implemented these suggestions.

- *I recommend the authors adjust verb tense and change tense from passive to active whenever possible to make sentences more concise. Examples provided below, but I would strongly recommend editing these throughout the manuscript.*

Comment: We agree.

Change: We removed passive expressions, wherever adequate.

Detailed commentsAbstract:

Line 54	No hyphen needed between whisker and density when used as a noun Comment & Change: Done.
Line 54-55	Please provide formal scientific names at first mention of the common names of the two elephant species tested Comment: We mention the scientific names of the three extant species early in the text, but not at the very beginning of the abstract. We mention them in the introduction, when we discuss species differences in trunk morphology. Change: None. We think the current order makes more sense than an earlier mentioning of the scientific names.
Line 56-61	The authors state that the two species show differences in whisker counts in Lines 54-55 but then describe elephants as a whole afterwards. It's unclear whether the statements apply to both species or just one, so I would recommend providing some kind of transition or clarifying phrase here. Comment: We agree. Change: We rewrote the abstract and made sure to specify whether results apply to just one or both species.
Line 62	"Whiskers show whisking" is hard to picture – I would recommend making the elephants the subjects of this sentence. Comment & Change: We agree and implemented this suggestion.
Line 52 & 65	I would suggest broader statements at the beginning and end to place the study and findings in a larger context for the field. Comment: As pointed out above, we agree with this suggestion. Change: We rewrote the abstract accordingly.

Introduction

Line 77	This is a confusing sentence that makes it sound like the innervation weighs 1.5 kg, but I think the author means the trunk? Either way, I recommend this be clarified. Comment: It is the innervation, i.e. the nerves and the trigeminal ganglia that weigh 1.5 kg. Change: We clarified this phrase.
---------	---

3

Line 78-80	This sentence could be clarified to make your point more clear - explain to the reader why does it matter that the nerve is thicker than the optic or vestibulococlear nerve. Based on the citation, it seems like you may want to make the point about thickness improving temporal precision of tactile signals. Comment & Change: We added an explanatory sentence here. (Page 3).
Line 80-81	“However” seems redundant after “Despite” Comment & Change: We agree and implemented this suggestion.
Line 99-102	This is an example of where sentence structure could be simplified to “Such morphological differences between African and Asian elephants match species-specific differences in trunk use—African elephants tend to pinch objects with their two fingers, and Asian elephants tend to grasp/wrap objects with their trunks.” Comment & Change: We implemented these suggestions.
Line 104	Morphological and behavioral specializations can be classified as “tactile” - I think you mean to contrast what is known about the trunk skin versus the trunk vibrissae to set up this paragraph. Comment & Change: The sentence mentioned by the referee was poorly worded and we rewrote it.
Line 113	What do you mean by “followed the pioneering work” – are you using the same methods? If so, I recommend you state that to be more specific. Comment & Change: We implemented this suggestion.
Line 114-119	These questions could be condensed to  1. How do the trunk whiskers of African and Asian elephants differ in density, distribution, morphology, and innervation? 2. How do similarities or differences affect function? Comment: We see the referee’s reasoning, but still believe that a more detailed spelling out of our questions is beneficial for clarifying the goals of our analysis. Change: None.
Line 124	A concluding sentence to place findings/study in broader context would be helpful here. Comment: We agree

	Change: We added a concluding sentence.
--	--

Methods

Line 131	I know you reference sex/number in Table 1, but it would be helpful to have a summary sentence here to state how many animals, the range of ages, and sexes in this sentence. Comment: As already pointed out in our earlier responses we added statistical detail and sample information throughout the ms. Change: We carefully implemented the sample size for each experiment throughout the manuscript.
Line 137	“insurmountable” seems grandiose – this could be simplified to something like “serious” Comment & Change: We implemented this suggestion.
Line 143	Would be helpful to have specifics on number of photographs taken/analyzed for each individual – you could add this to the text and to Table 1. Comment: We agree. Change: We specified the number of pictures counted for each sample.
Line 144-145	Needs clarification - as is, it sounds like some of the trunk tips were not fixed or frozen. What happened to the others? You could state, “Out of the XXX trunk tips photographed, XX were fixed and XX were frozen for further analysis.” Comment & Change: We clarified this sentence.
Line 145	For counting what? Please be specific. Comment & Change: We added this info.

Line 149-151	Were counts performed by the same human or different humans? If the latter, how did you assess inter-observer reliability? Comment & change: The counts were performed from one the authors. We agree that the inter-observer reliability is an important factor to consider for the whisker count. We let a coauthor do a recount on two of the samples with a deviation of 5% and 11% in both directions.
Line 149	Here is a place you could specify sample size – “For every specimen (n = XX)” Comment & Change: We added this info.

Line 152	I recommend that “baby elephant” should be referred to as “calf” throughout the manuscript, but I recognize that published work exists using this phrase, so happy to leave this up to the authors’ discretion. Comment: We agree that the term baby sounds colloquial. Change: We replaced the term baby with newborn throughout the ms.
Line 155-159	Sampling needs to be clarified for whisker thicknesses” You state the “for each specimen” then say that this was only performed on two adults from each species. To clarify, you can state “ For two adults from each species, the thicknesses of ten whiskers were measured...” Comment & Change: We agree and added this info.
Line 162-171	Please provide details on the number of behavioral trials you completed for each individual, and thus, videos that you analyzed to assess behavior. Comment: The whiskers did not move in all high-quality videos of fruit retrieval ($n > 40$ from experiments with one individual). We verified this visual impression in $n = 5$ trials with quantitative tracking. This analysis included whiskers from both sides of the trunk. Change: We included the number of videoclips and the number of whiskers we analyzed in the corresponding method section.
Line 176	How many whiskers were tracked for each animal and how were they chosen? Did you examine whiskers from different sides of the trunk? Comment & Change: See our response above.
Line 165-168	This is an example of sentence that can be simplified to improve clarity: “We observed adult Asian elephants during routine training ($n = 5$) and grasping behaviors based on haptic discrimination ($n=1$). For the latter, a 10-year-old female, Anchali, retrieved fruit...” Comment & Change: We agree and implemented this suggestion.
Line 186-187	You don’t need to bring the figure up here, especially since it’s not the first one. Instead, you can state that “We identified the onset of suction for apple vacuuming using the video audio track.” Comment & Change: We agree and adapted the sentence.
Line 189	Please provide more details on the sample sizes for whole trunks and trunk tips. Comment & Change: We provided details on the ns at the end of the paragraph.

Line 190-191	Methods are unclear here - what do you mean specifically by FSCs were dissected or processed as a whole? Do you mean they were cut into; if so, how were they cut into and at what plane? Or, do you mean that they were separated from the surrounding tissue, in contrast to the being processed while still embedded in tissue? Please be detailed. Comment & Change: We rewrote the sentence to make the method clear.
Line 214	Please provide more details on the sample sizes for trunks. Comment & Change: We implemented detailed information about the number of samples throughout the paragraph.
Line 225	Please specify if this was a facial whisker pad. Comment & Change: We implemented this information.
Line 238-240	The authors must provide more information on the statistical tests they used, any relevant assumption checks to assess significant differences between species, and how the data + error are reported (ex. means + sem). Comment & Change: The referee is right, as already pointed out above, we added our statistical testing procedure in the corresponding method section
Line 251	Gentle reminder that data should be plural. You use it correctly in the line

	before, so just carry that over into this sentence. Comment & Change: The referee is right, we corrected the sentence.
--	--

Results

	Great use of headers throughout the results! I would recommend adjusting the following to include the result in the statement, similar to your other headers in this section: "Elephant trunk whisker thickness and geometry", "Anatomy and morphological variety of trunk whisker follicles", "Ventral trunk ridge whisker arrays and their behavioral contact patterns" Comment & Change: A great suggestion that makes the results read more cohesive.
Line 260	What do you mean by more prominent? Longer, thicker, etc? Please describe this with regard to the attributes of the whiskers. Based on the figure from the side view it does seem that they are longer, although they also look darker (which would make them appear more prominent on the white background). Comment: The whisker of African elephants are more obvious to the observer, a property, to which we refer as 'prominent'. Change: None.

Line 269- 270	This is the first mention of the “fingertip” which I take to mean the peripheral-most part of the tip? I would recommend defining this more explicitly when you first mention it as an anatomical feature. Also, you mention that they pinch objects with the fingertip, but this seems like it will become more of a Discussion point than a Results point unless you are referring to specific observations of elephants included in this study. Comment: We agree. Change: We define fingertip in the revised ms.
Line 270- 272	The authors show whisker density in Fig 1e/f but then compare absolute number of whiskers in Fig 1g. The absolute numbers are difficult to interpret, as it’s not clear how this accounts for any differences in surface area between the species. The authors should report density, in addition to absolute counts, and run the analysis on density to interpret the observed differences. Comment: We understand the referee’s reasoning behind this comment. We provided whisker density information for n = 2 trunk tips already in our original submission (Figure 1e,f). As can be observed in these panels whisker density varies strongly across trunk tip. We therefore think that overall averages of whisker density will provide the reader only with little additional information. Change: None.
Line 275	Similar to my comment in Line 260 - it’s not clear to me what “prominent” means. I recommend the authors explicitly state what they mean by this and ensure that it coincides with statistical analyses. Comment: Please see our response above.
Line 279	“Accordingly” seems to be an unnecessary transition here. Comment & Change: We agree and rewrote this sentence
Line 280- 282	The word “such” seems unnecessary and a bit jarring in these sentences. Comment & Change: We agree and replaced such by this.
Line 291	I recommend the authors be more explicit here. Rather than “showed whisker length lateralization that is expected from such behavior...” please state what that lateralization actually was. Comment & Change: We agree and explain the connection by behavioral lateralization and abrasion in the revised text.
Line 295- 298	The way this is phrased makes it sound like you are comparing your results to those in rats from other studies based on this phrase: “an observation that is common in rats”. If you are comparing your results to the rats you examined in this study, then please make it clear. If you are comparing your results to the rats examined in other studies, then this comment would fit better in the Discussion. Comment & Change: We agree and added the expression ‘in other studies on rats’.
Line 299- 300	This sounds too speculative, which is odd to include in Results, unless you actually measured whisker growth rates in elephants. Comment & Change: We removed the speculative sentence.

Line 317- 318	Similar to my comment about “prominent” - thick and sturdy seems open to interpretation. I would recommend concluding that the whiskers have a cylindrical shape and are thicker than rats in both species, and variable thickness across the trunk tip in African elephants. Comment: We understand the reasoning behind the referee’s criticism. Still, we think that descriptive adjectives such prominent, thick and sturdy allow the reader to form more clear impression of elephant whiskers. Change: None.
Line 322- 340	Are the microCT scans in this section different than the ones mentioned in the previous section (Elephant trunk whisker thickness and geometry)? The authors mention the rat in the previous section but focus that section on elephants. I actually wonder if including the rat microCT is necessary at all - the only result the authors discuss from it is that the elephant follicles are much larger. This doesn’t seem to be a super informative result since it may be confounded by body and appendage size. If the authors have an H&E-stained whisker follicle, that may be helpful to include, especially since the authors make a note that the elephants lack vibrissal capsular muscles that rats have. If not, I would recommend either (1) discussing the similarities/differences between the microCT scans of the rat and elephant beyond absolute size, or (2) removing the rat microCT from the results. Comment: We think that the side-by-side comparison of elephant and rat follicle is instructive for the reader, in particular for the large community of scientists familiar with rodent whiskers. Change: None.
Line 328	When mentioning other species in this line, the authors should provide references. In contrast to my point below, I can see how this is a Results point because the authors are using a reference to other species to indicate “typical” FSC structure. Comment: We agree. Change: We added references that include the typical FSC structure of other species (mouse, rat, hyrax).
Line 333- 345	“seen in other species, such as rodents” seems to be a Discussion point unless the authors have an H&E-stained section of a rodent whisker from this study they can add to the figure to point out the differences. Comment: We agree. Change: We removed the reference to other species.
Line 339- 340	The authors provide information about the statistical test in the figure caption, but this information should also be provided in text when stating significance. Comment & Change: We agree and added the statistical information also in the text.

Line 341- 342	These summarized results don't seem to match the text. Being large is a relative term and likely related to body size, appendage size, and whisker geometry/length. To me, it seems the results of this section are (1) elephant FSCs in adults and calves have typical structure and lack vibrissal capsule muscles, and (2) show differences in length according to trunk region in calves (the pattern may be nonexistent or different in adults, but since the authors didn't examine that, the calf distinction is important here). Comment: We agree. Change: We added a qualifier to our statement about elephant whiskers being large.
Line 344	These results seem to be from one elephant calf, no? If so, the header should be clarified that calf elephant trunk whiskers are densely innervated. The density and distribution of innervation may change with maturation, so it's important to clarify the context of your results. Comment: We looked at the innervation of both adult and newborn elephant FSCs, but only in the investigated newborn the quality of the staining was sufficient for axon counts, so the count is only derived from the newborn. Change: We adjusted the headline of the paragraph.

7

Line 345	“Comparable to FSCs of other species” feels like a discussion point as it is used here. Comment: We agree. Change: We removed this phrase.
Line 358- 460	This should be moved to the methods with clarification on whether axon numbers were counted in one individual (and age) or multiple individuals. Comment: We agree. Change: We added this info and moved the sentence.
Line 376- 402	All parts of this section that are not results should be moved to the Methods section “Videography and whisker tracking”, along with more clarification on how many whiskers were tracked for the individual. The methods should not be repeated here. Line 396-397 includes what seems like a Discussion point. Comment: We disagree and think that the findings described here are difficult to understand without the direct methodological context. Change: None.
Line 405- 407	What was the surface area of the trunks? That seems important to scale the absolute number of whiskers. Comment & change: We do not have this information for many of our specimens and this data is also not easily derived from photographs. A rough idea of trunk surface area can be deduced from Figure 7a-d, where we show photographs of both the dorsal and ventral side of newborn elephant trunks.

Line 414	I'm not sure I agree with the statement that whiskers are distributed more evenly on the dorsal trunk side. Although there isn't a distinct two-row pattern as in the ventral trunk, there still seems to be a pattern, as the authors, note from the peripheral to the proximal trunk (although the pattern is different between the species). Comment: This comment is a difficult call for us. We do not disagree with the referee that there might be some patterning in dorsal trunk whiskers. Still, the pattern is not entirely clear and our expression 'more evenly' was a deliberate attempt to stay neutral and not to overinterpret our observations. Change: None.
Line 419-429	Much of these lines should be moved to the Discussion. You should mention that you personally observed and photographed elephant grasping behaviors (I'm assuming the photos in e-f are the authors'), but make sure to include that in the Methods, as well. Comment & Change: We agree that we report interpretations and methodological detail in this section. Still, this information is helpful in understanding the results presented here and this is why we would like to keep it in place.
Line 428-429	"Impressive whisker density" is hard to interpret. The ventral ridge appears to have a much higher density than the rest of the trunk, which coincides with observations from the microCT scan. Whether or not this is impressive is subjective. Comment: We agree. Change: We rewrote this sentence.
Line 434-435	The main conclusion seems to be that whisker density varies across the trunk region in elephant calves, which seems to align with observed adult elephant grasping behaviors. The context of age is important, since whisker density and behavior may vary across development and as the trunk grows. Concluding that the whisker arrays serve for object sensing/balancing is more of a Discussion point. Keep the results strictly to the results rather than speculation on broader function. Comment: We agree. Change: We altered the conclusion sentence to a more neutral statement.

Discussion

Line 441	"as a result of lateralized trunk behavior" – this would be more of a correlation or association, no? Likely associated based on the authors' observations but not definitive as a result unless manipulated experimentally. Comment: We agree that this sentence was worded too strongly given our limited experimental evidence. Change: We added 'probably' as a qualifier.
----------	--

Line 455	This line should have a citation. Comment: We state here that there is a lack of evidence, and hence it is difficult to come up with a specific reference.
Line 479	This first sentence needs a citation. Comment: We agree. Change: We added a reference.
Line 480	I would recommend replacing “master tusk” with “dominant tusk” or “primary tusk”. Comment: We agree that dominant tusk would seem more appropriate. Change: We did not implement this change, because we wanted to maintain the terminology from the literature.
Line 488-490	I’m not convinced that this difference in follicle length is as striking when considering the vastly different sizes of elephants and rodents. I recommend the authors focus more on the internal structure and innervation rather than absolute size. Comment & Change: We agree and now mention the difference in body size as a potential explanation for follicle size.
Line 486-518	I found this section a bit difficult to follow. Comment: We understand this criticism. Change: We removed two complicated sentences and also added an explanatory sentence. We hope the section reads better now.
Line 523-535	Please expand on this hypothesis more – what are “mixed food preferences” vs. “grazing/browsing” and how would you expect that to influence the differences you observed? Comment: We agree. Change: We added a sentence explaining our reasoning.
Line 532-534	Manatees and elephants have quite different habitats, which many partially explain the thickness differences. Other aquatic mammals have thick whiskers (seals, sea lions, sea otters) compared to terrestrial closely-related species. Comment: The referee is right. This critique aligns with a similar remark of referee 1 and we have addressed it there, please see our response above.
Line 547-548	This statement needs a citation, as well as further explanation for how this influences whisker mobility. It’s a compelling point, but a stronger statement would help this shine through. Comment: We agree. Change: We added a sentence explaining our reasoning.
Line 554	What do you mean by “more elaborate”? Comment & Change: We replaced elaborate by conspicuous, a more descriptive terminology.

Tables

Table 1	I wonder if this table could be a good place to help clarify sample sizes for different data streams. This table would be strengthened by including a metric of body size or trunk size (if known) to help scale the whisker counts, a column for lateralization in whiskers observed, as well as include the axon numbers per FSC counted for the grayed individuals used for FSC dissection. The authors could also consider ordering the individuals by age within the species grouping, which would help any patterns by age to stand out more. Comment: As the whisker count was only performed on one newborn elephant we don't think adding another column would add clarity. We also can't provide information about body or (whole) trunk size for many of the samples. Change: None.
---------	--

Figures (need to check figure captions)

Fig. 1	Beautiful figure layout and photos! You provide an description of the inset showing whiskers for a but not b – please Comment & Change: We are thankful for the appreciation and added the missing information.
--------	---

	clarify, as you do in the text, that both insets show whiskers. It may be helpful to label the pinching tip
Fig. 2	e, use of “respectively” doesn’t make sense to me here. I also found the x-axes confusing here and had to look at them awhile to understand them. Perhaps remove “longer whiskers” from the bottom x-axis? Comment & Change: We removed respectively.
Fig. 3	e. would be helpful to draw box around trunk tip and ventral, as you do for dorsal and lateral to make e and f parallel Comment: The pictures in e are chosen to provide a visual thickness comparison of dorsal and lateral whiskers in an African elephant. We think adding a box for the other regions would confuse the reader as they don’t correspond with the magnified insets above. Change: We added the description ‘dorsal’ and ‘lateral’ on the magnified sections to emphasize that these correspond to the marked regions in the lower part of the figure.
Fig. 4	d-g: reading the white text on these figures is challenging - I can see why the authors chose white, but perhaps you could play with bolding the acronyms to make them slightly more clear? Comment: We agree. Change: We bolded the text to improve visibility. d & f: I would recommend providing context in the caption for the different whisker lengths. It looks like d was cut off during sampling and f is either growing in or has broken off? Providing a quick explanation would be helpful since it is an obvious difference that I’m assuming is not an actual result but rather an artifact. Comment: We agree. Change: We added an explanation to the legend of panel f. d-e: I would recommend making the VSS lines more clear; it’s a bit hard to see where the line is pointing. Comment: We agree. Change: We changed the thickness and placement of the line. f-g I would recommend labelling these figures with acronyms to point out structures the same way you do for d-e. If not all the structures are evident in the figure due to slicing artifacts, you can mention that in the caption. Comment: We agree that this improves comparability Change: We labeled the figures accordingly for all structures evident on the respective micrographs.
Fig. 5	a-c, e-f: I would strongly recommend adding arrows/symbols to help direct the reader to important areas of these figures. Comment: We agree that arrows in e and f would add clarity. As a and b should provide an overview of a typical section rather than emphasizing a specific area, we didn’t add any arrows there. Change: We added arrows in e and f for emphasizing important structures.

	D: This caption needs more information about the number of elephants these data represent. Comment: As only a few of our samples had sufficient tissue quality to be used for counting the axons from immunohistological stainings the axon count was restricted to one baby Asian elephant. Change: We added this information in the figure caption as well as the methods and the corresponding results paragraph.
Fig. 6	I would recommend adding “vibrissae” to “elephants do not whisk” as in “elephant vibrissae do not whisk” to the first line and providing an arrow for the tracked whisker in c and g to match the arrows in d and h. Comment & Change: We agree and changed the heading accordingly. Additionally, we added the arrows in c and g.

Reviewer #3 (Remarks to the Author):

The paper by Deiringer et al. reports characteristics of whiskers of elephant trunks. They offer fundamental information about trunk whiskers with valid methodology, and I was surprised that they had not been studied before this study (this was well reflected in the simple, concise title of this study). Therefore, I believe this study is important to better understand not only elephant behavior but also inter-species diversity of how whisker anatomy shapes mammalian behavior. They set clear questions to be addressed in this study, which makes readers easily understand their points. The strength of this study is that they well-integrate results from both anatomical and behavioral aspects. For example, the asymmetry of the whisker trunk length is noteworthy. They give important insights that trunk whisker does not whisk, unlike rats. Their argument that elephants have dramatically improved trunk mobility over the course of evolution and therefore do not need specific mobility of whiskers is very interesting. They also give a next question about how whisker grows in elephants, which would facilitate further research to follow. Overall, their findings about elephant whiskers, which vary according to region, age, species, and behavior will give us fundamental new insights to better understand elephants. Whisker studies in mammals are highly biased in rodents, and therefore this study has an important piece to fill the fundamental gap. I only have some minor comments.

Comment: The referee summarizes our results and notes that it is surprising that we do not already know about a range of findings that we document. This assessment resonates with the motivation of our study, name that the tactile biology of elephants has been neglected. The referee appears to appreciate our work and we are thankful for this positive assessment.

Change: None.

Specific comments

1. Line 269: *The statement here seems different from that in the introduction, where you state that “while African elephants tend to pinch objects with their two fingers, Asian elephants tend to grasp/wrap objects with their trunk. Please clarify this point.*

Comment: We agree that there is an apparent contradiction in the terms that were used.

Change: We added an explanatory qualifier in this sentence.

2. Line 340: *Figure 4i. Is this data from African or Asian elephants? Please clarify this in the figure legend. This comment is applied to Figure 5d.*

Comment & Change: The referee is right, this information was missing and we added it.

Reviewer #4 (Remarks to the Author):

Please see the attached annotated MS word document of the manuscript.

Comment & Change: We implemented the suggested changes and highlighted the changes in the revised ms.

Thank you for allowing me to review “Elephant Trunk Whiskers.” I find the topic and scope to be excellent and appropriate for Communications Biology. The data provided in this manuscript is excellent and the ecological interpretations are clear and supported (mostly).*

Comment: The referee finds our topic and scope fitting and is overall very positive about the ms. We are encouraged by these comments.

Change: The referee adds the qualifier mostly, which we address below when dealing with the more detailed comments.

Throughout the manuscript review attached, I have provided suggestions on how to improve the clarity and flow of the document. Small additions like “define what a vibrissa is and what its key features (i.e., sinus?) are” may help assist future readers to navigate this interesting paper.

Comment: These detailed comments offer a trove of insights and we think that our ms greatly improved by the revisions that followed from these comments. We are thankful for the hard work that the referee put in here.

Change: All these comments were addressed point-by-point.

Overall, I found this manuscript to be very enjoyable, and left me wanting to know more about the histomorphology of the elephant trunk. This spark of curiosity is a direct result of the author's ability to show patterns between form and function.

Comment: We put in an effort to show as much of the trunk as possible; we are glad that the referee enjoyed our style of reporting.

Change: None.

I did have some questions about how the behavioral data and the “rat” data were included in this paper – as they were not outlined in the methods. Not included in my “comments” in the MS WORD document provided – was a question about body size and scaling. I wonder if you have found any size-related variation? Interestingly, I have found some size-related differences in vibrissa density across mammals and I have seen age-related differences in mechanoreceptor density as well. I would have liked to have seen a small comment about the size of these two elephant populations. I fully appreciate that there are negligible differences in size between African and Asian elephants, but when you include stillborn, “baby,” subadult, and adult specimens in a study – SIZE is a question that immediately comes to mind.

Comment & Change: In this broad comment the referee notes potential problems with our sample. We tried to address this concern throughout the ms by providing more detailed ns and statistical detail (as also requested by the other referees).

The discussion was amazingly clear and straightforward. However, a small concluding statement about the observed variation might be informative (directions for future work, e.g., histomorphology of the non-hairy (glabrous?) ventral surface of the trunk).

I think that adding a concluding statement may be helpful for readers like myself who wanted to know more, but assumed the authors did not collect the needed data to address my questions. However, the authors did present several hypotheses throughout – that could be “re-explored” or “re-stated” in a concluding statement.*

Comment: The conclusion needs to be rewritten as we already agreed. Adding open questions to the conclusion is an interesting idea.

Change: We rewrote the conclusions para.

In the end, I am very impressed with the author's holistic approach to comparative functional anatomy and I very much enjoyed reading it.

Comment: Another very appreciative comment, which we read with pleasure.

Change: None.

REVIEWERS' COMMENTS:

Reviewer #1 (Remarks to the Author):

Thank you for re-writing the manuscript and addressing all my comments. I really feel that the story is much clearer and better justified now.

Just a couple of minor things I observed:

1. Throughout, just give everything one last read-through to make sure it is all as clear as it can be, especially tenses.
2. In the abstract, please do make clear what is background information and what you have found in your study, this was not overly clear.
3. In Intro: R. Sprinz should just be Sprinz?

I would be happy to see this publication published and add to your growing research work on elephant whiskers. Well done.

Reviewer #2 (Remarks to the Author):

Deiringer and colleagues have beautifully revised this manuscript, and I do not have many comments at all. The revisions allow the reader to more easily follow the methods, sample sizes, and interpret the results in a larger comparative, ecological context.

I still recommend the authors develop a stronger title that provides information about the species and the methods. Doing this will emphasize that the authors combine structure and function in their approach to describing the whiskers in these two species.

Fig. 2e - I still think the addition of "longer whiskers" on the x-axis of the newborn graph is confusing. Aren't both graphs showing a comparison between right longer whiskers, symmetric, and left longer whiskers - the only difference between the graphs is age? I recommend clarifying if it's important to the authors to retain this phrase in one of the graphs but not the other.

Below, we repeat the referee comments (in italics) before responding to them.

Reviewer #1:

Thank you for re-writing the manuscript and addressing all my comments. I really feel that the story is much clearer and better justified now. Just a couple of minor things I observed:

1. Throughout, just give everything one last read-through to make sure it is all as clear as it can be, especially tenses.

We looked at tenses throughout the ms, but found them to be adequate in almost all cases.

2. In the abstract, please do make clear what is background information and what you have found in your study, this was not overly clear.

Comment & Change: We agree and we added a 'the following findings' phrase to clarify that point.

3. In Intro: R. Sprinz should just be Sprinz?

Comment & Change: The referee is right, we adjusted the sentence.

I would be happy to see this publication published and add to your growing research work on elephant whiskers. Well done.

Reviewer #2:

Deiringer and colleagues have beautifully revised this manuscript, and I do not have many comments at all. The revisions allow the reader to more easily follow the methods, sample sizes, and interpret the results in a larger comparative, ecological context.

Comment: We are thankful for the positive assessment.

Change: None.

I still recommend the authors develop a stronger title that provides information about the species and the methods. Doing this will emphasize that the authors combine structure and function in their approach to describing the whiskers in these two species.

Comment & Change: We implemented this suggestion.

Fig. 2e - I still think the addition of 'longer whiskers' on the x-axis of the newborn graph is confusing. Aren't both graphs showing a comparison between right longer whiskers, symmetric, and left longer whiskers? the only difference between the graphs is age? I recommend clarifying if it's important to the authors to retain this phrase in one of the graphs but not the other.

Comment & Change: We added the phrase to the other plot as well because I feel like it is not obvious what is even shown without that phrase.